# BitsFusion: 1.99 bits Weight Quantization of Diffusion Model

**Yang Sui**[1,2,†]     **Yanyu Li**[1]     **Anil Kag**[1]     **Yerlan Idelbayev**[1]     **Junli Cao**[1]     **Ju Hu**[1]
**Dhritiman Sagar**[1]     **Bo Yuan**[2]     **Sergey Tulyakov**[1]     **Jian Ren**[1,*]
[1]Snap Inc.     [2]Rutgers University
Project Page: https://snap-research.github.io/BitsFusion

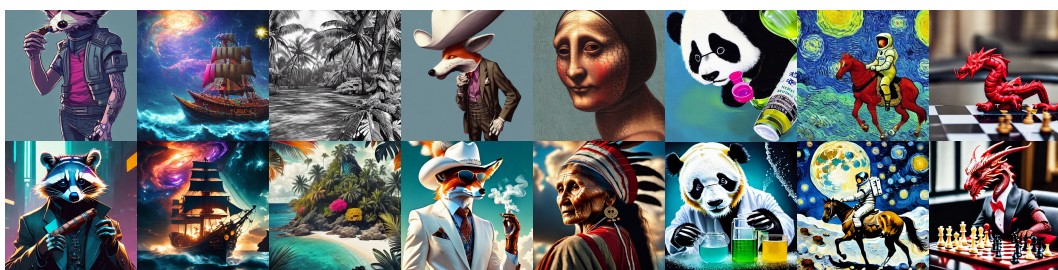

Figure 1: *Top*: Images generated from full-precision Stable Diffusion v1.5. ***Bottom***: Images generated from BitsFusion, where the weights of UNet are quantized into *1.99* bits, achieving $7.9\times$ *smaller* storage than the one from Stable Diffusion v1.5. All the images are synthesized under the setting of using PNDM sampler [49] with 50 sampling steps and random seed as 1024. Prompts and more generations are provided in App. M.

## Abstract

Diffusion-based image generation models have achieved great success in recent years by showing the capability of synthesizing high-quality content. However, these models contain a huge number of parameters, resulting in a significantly large model size. Saving and transferring them is a major bottleneck for various applications, especially those running on resource-constrained devices. In this work, we develop a novel weight quantization method that quantizes the UNet from Stable Diffusion v1.5 to 1.99 bits, achieving a model with $7.9\times$ smaller size while exhibiting even *better* generation quality than the original one. Our approach includes several novel techniques, such as assigning optimal bits to each layer, initializing the quantized model for better performance, and improving the training strategy to dramatically reduce quantization error. Furthermore, we extensively evaluate our quantized model across various benchmark datasets and through human evaluation to demonstrate its superior generation quality.

## 1 Introduction

Recent efforts in developing diffusion-based image generation models [77, 31, 79, 21, 80] have demonstrated remarkable results in synthesizing high-fidelity and photo-realistic images, leading to various applications such as content creation and editing [68, 67, 61, 71, 90, 88, 50, 40], video generation [20, 75, 3, 1, 16, 57, 15], and 3D asset synthesis [87, 44, 64, 74, 65], among others.

---

[†]Work done during an internship at Snap Inc.
[*]Corresponding author.

38th Conference on Neural Information Processing Systems (NeurIPS 2024).

However, Diffusion Models (DMs) come with the drawback of a large number of parameters, *e.g.*, millions or even billions, causing significant burdens for transferring and storing due to the bulky model size, especially on resource-constrained hardware such as mobile and wearable devices.

Existing studies have explored reducing the model size of large-scale text-to-image diffusion models by designing efficient architectures and network pruning [41, 92, 32]. These approaches usually require significant amounts of training due to the changes made to the pre-trained networks. Another promising direction for model storage reduction is quantization [12, 30], where floating-point weights are converted to low-bit fixed-point representations, thereby saving computation memory and storage.

There have been emerging efforts on compressing the DMs through quantization [73, 38, 17, 39]. However, these approaches still face several major challenges, especially when quantizing large-scale text-to-image diffusion models like Stable Diffusion v1.5 (SD-v1.5) [70]. *First*, many of these methods are developed on relatively small-scale DMs trained on constrained datasets. For example, models trained on CIFAR-10 require modest storage of around 100 MB [21, 39]. In contrast, SD-v1.5 necessitates $3.44$ GB of storage in a full-precision format. Adapting these methods to SD-v1.5 remains to be a challenging problem. *Second*, current arts mainly focus on quantizing weights to $4$ bits. How to quantize the model to extremely low bit is not well studied. *Third*, there is a lack of fair and extensive evaluation of how quantization methods perform on large-scale DMs, *i.e.*, SD-v1.5.

To tackle the above challenges, this work proposes BitsFusion, a quantization-aware training framework that employs a series of novel techniques to compress the weights of pre-trained large-scale DMs into *extremely low* bits (*i.e.*, 1.99 bits), achieving even *better* performance (*i.e.*, higher image quality and better text-image alignment). Consequently, we compress the $1.72$ GB UNet (FP16)[1] of SD-v1.5 into a 219 MB model, achieving a $7.9\times$ compression ratio. Specifically, our contributions can be summarized into the following four dimensions:

- **Mixed-Precision Quantization for DMs.** We propose an effective approach for quantizing DMs in a mixed-precision manner. *First*, we thoroughly analyze the appropriate metrics to understand the quantization error in the quantized DMs (Sec. 3.2). *Second*, based on the analysis, we quantize different layers into different bits according to their quantization error (Sec. 3.3).
- **Initialization for Quantized DMs.** We introduce several techniques to initialize the quantized model to improve performance, including time embedding pre-computing and caching, adding balance integer, and alternating optimization for scaling factor initialization (Sec. 4.1).
- **Improved Training Pipeline for Quantized DMs.** We improve the training pipeline for the quantized model with the proposed two-stage training approach (Sec. 4.2). In the first stage, we use the full-precision model as a teacher to train the quantized model through distillation. Our distillation loss forces the quantized model to learn both the predicted noise and the intermediate features from the teacher network. Furthermore, we adjust the distribution of time step sampling during training, such that the time steps causing larger quantization errors are sampled more frequently. In the second stage, we fine-tune the model using vanilla noise prediction [21].
- **Extensive Quantitative Evaluation.** For the first time in the literature, we conduct extensive quantitative analysis to compare the performance of the quantized model against the original SD-v1.5. We include results on various benchmark datasets, *i.e.*, TIFA [25], GenEval [13], CLIP score [66] and FID [19] on MS-COCO 2014 validation set [46]. Additionally, we perform human evaluation on PartiPrompts [86]. Our 1.99-bit weights quantized model *consistently outperforms* the full-precision model across various evaluations, demonstrating the effectiveness of our approach.

## 2 Related Works

To enhance model efficiency in terms of storage and computational costs, quantization [11, 59, 58, 43, 36, 60, 48, 84, 45, 53] is adopted for diffusion models [73, 38, 18, 76, 81, 83, 51, 85, 4, 82, 7, 93, 27, 17, 39, 91] with primarily two types: post-training quantization (PTQ) and quantization-aware training (QAT). PTQ does not require a full training loop; instead, it utilizes a limited calibration dataset to adjust the quantization parameters. For example, PTQ4DM [73] calibrates the quantization parameters to minimize the quantization error of DMs. Q-Diffusion [38] minimizes the quantization error via the block-wise reconstruction [42]. PTQD [18] integrates quantization noise into the

---

[1]For SD-v1.5, we measure the generation quality using the FP32 format. However, since SD-v1.5 FP16 has similar performance to SD-v1.5 FP32, we use SD-v1.5 FP16 to calculate our compression ratio.

stochastic noise inherent in the sampling steps of DMs. TDQ [76] optimizes scaling factors for activations across different time steps, applicable to both PTQ and QAT strategies. TFMQ [27] focuses on reconstructing time embedding and projection layers to prevent over-fitting. However, PTQ often results in performance degradation compared to QAT, particularly when aiming for extremely low-bit DMs. In contrast, QAT involves training the full weights to minimize the quantization error, thereby achieving higher performance compared to PTQ. For instance, EfficientDM [17], inspired by LoRA [24], introduces a quantization-aware low-rank adapter to update the LoRA weights, avoiding training entire weights. Q-DM [39] employs normalization and smoothing operation on attention features through proposed Q-attention blocks, enhancing quantization performance. Nevertheless, existing works primarily study $4$ bits and above quantization on small-scale DMs trained on constrained datasets. In this paper, we focus on quantizing large-scale Stable Diffusion to extremely low bits and extensively evaluating the performance across different benchmark datasets.

# 3 Mixed Precision Quantization for Diffusion Models

In this section, we first go through the formulations of weight quantization and generative diffusion models. We then determine the mixed-precision strategy, assigning optimized bit widths to different layers to reduce the overall quantization error. Specifically, we first analyze the quantization error of each layer in the diffusion model and conclude sensitivity properties. Then, based on the analysis, we assign appropriate bits to each layer by jointly considering parameter efficiency (*i.e.*, size savings).

## 3.1 Preliminaries

**Quantization** is a popular and commonly used technique to reduce model size. While many quantization forms exist, we focus on uniform quantization, where full-precision values are mapped into discrete integer values as follows:

$$\boldsymbol{\theta}_{\texttt{int}} = \texttt{Clip}(\lfloor \frac{\boldsymbol{\theta}_{\texttt{fp}}}{\mathbf{s}} \rceil + I_z, 0, 2^b - 1), \tag{1}$$

where $\boldsymbol{\theta}_{\texttt{fp}}$ denotes the floating-point weights, $\boldsymbol{\theta}_{\texttt{int}}$ is the quantized integer weights, $\mathbf{s}$ is the scaling factor, $I_z$ is the zero point, and $b$ is the quantization bit-width. $\lfloor \cdot \rceil$ denotes the nearest rounding operation and $\texttt{Clip}(\cdot)$ denotes the clipping operation that constrains $\boldsymbol{\theta}_{\texttt{int}}$ within the target range. Following the common settings [38, 17], we apply the channel-wise quantization and set $8$ bits for the first and last convolutional layer of the UNet.

**Stable Diffusion.** Denoising diffusion probabilistic models [77, 21] learn to predict real data distribution $\mathbf{x} \sim p_{\text{data}}$ by reversing the ODE flow. Specifically, given a noisy data sample $\mathbf{z}_t = \alpha_t \mathbf{x} + \sigma_t \boldsymbol{\epsilon}$ ($\alpha_t$ and $\sigma_t$ are SNR schedules and $\boldsymbol{\epsilon}$ is the added ground-truth noise), and a *quantized* denoising model $\hat{\boldsymbol{\epsilon}}_{\boldsymbol{\theta}_{\texttt{int},s}}$ parameterized by $\boldsymbol{\theta}_{\texttt{int}}$ and $\mathbf{s}$, the learning objective can be formulated as follows,

$$\mathcal{L}_{\boldsymbol{\theta}_{\texttt{int}},\mathbf{s}} = \mathbb{E}_{t,\mathbf{x}} \left[ \| \boldsymbol{\epsilon} - \hat{\boldsymbol{\epsilon}}_{\boldsymbol{\theta}_{\texttt{int}},\mathbf{s}}(t, \mathbf{z}_t, \mathbf{c}) \| \right], \tag{2}$$

where $t$ is the sampled time step and $\mathbf{c}$ is the input condition (*e.g.*, text embedding). Note that during the training of quantized model, we optimize $\boldsymbol{\theta}_{\texttt{fp}}$ and $\mathbf{s}$ by backpropagating $\mathcal{L}_{\boldsymbol{\theta}_{\texttt{int}},\mathbf{s}}$ via Straight-Through Estimator (STE) [2] and quantize the weights to the integers for deployment. Here, for the notation simplicity, we directly use $\boldsymbol{\theta}_{\texttt{int}}$ to represent the optimized weights in the quantized models.

The latent diffusion model [70] such as Stable Diffusion conducts the denoising process in the latent space encoded by variational autoencoder (VAE) [34, 69], where the diffusion model is the UNet [9]. This work mainly studies the quantization for the UNet model, given it is the major bottleneck for the storage and runtime of the Stable Diffusion [41]. During the inference time, *classifier-free guidance* (CFG) [22] is usually applied to improve the generation,

$$\tilde{\boldsymbol{\epsilon}}_{\boldsymbol{\theta}_{\texttt{int}},\mathbf{s}}(t, \mathbf{z}_t, \mathbf{c}) = w\hat{\boldsymbol{\epsilon}}_{\boldsymbol{\theta}_{\texttt{int}},\mathbf{s}}(t, \mathbf{z}_t, \mathbf{c}) - (w-1)\hat{\boldsymbol{\epsilon}}_{\boldsymbol{\theta}_{\texttt{int}},\mathbf{s}}(t, \mathbf{z}_t, \varnothing), \tag{3}$$

where $w \geq 1$ and $\hat{\boldsymbol{\epsilon}}_{\boldsymbol{\theta}_{\texttt{int}},\mathbf{s}}(t, \mathbf{z}_t, \varnothing)$ denotes the generation conditioned on the null text prompt $\varnothing$.

## 3.2 Per-Layer Quantization Error Analysis

**Obtaining Quantized Models.** We first perform a per-layer sensitivity analysis for the diffusion model. Specifically, given a pre-trained full-precision diffusion model, we quantize *each* layer to $1$, $2$, and $3$ bits while freezing others at full-precision, and performing quantization-aware training (QAT)

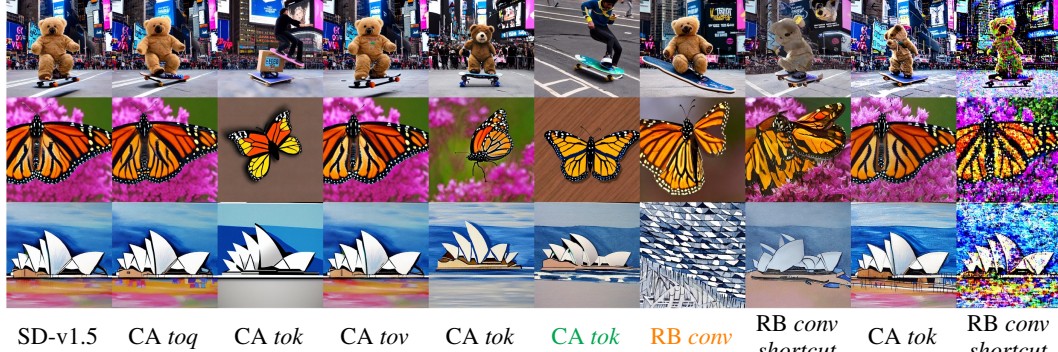

| SD-v1.5 | CA *toq* | CA *tok* | CA *tov* | CA *tok* | CA *tok* | RB *conv* | RB *conv* shortcut | CA *tok* | RB *conv* shortcut |

(a) Left most column shows the images synthesized by SD-v1.5 FP32 and other columns show images generated by the quantized models, where only one layer is quantized (*e.g.*, CA *toq* denotes the cross-attention layer for Query projection is quantized and RB *conv shortcut* denotes the Convolution Shotcut layer in Residual Block is quantized. The quantized layers follow *the same order* of highlighted layers in (b) and (c), from left to right. Quantizing the layers impact both the *image quality* (as in RB *conv shortcut*) and *text-image alignment* (*e.g.*, the teddy bear disappears after quantizing some CA *tok* layers).

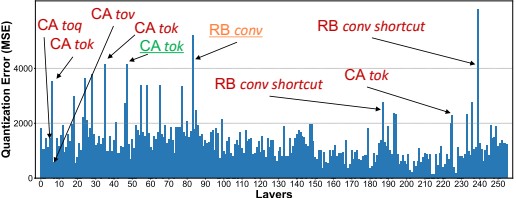

(b) MSE value by quantizing layers in SD-v1.5.

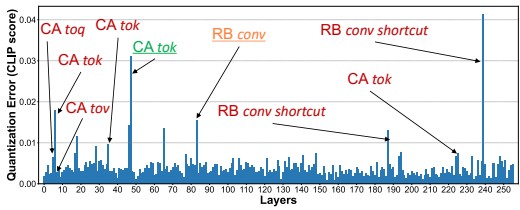

(c) CLIP score drop by quantizing layers in SD-v1.5.

Figure 2: 1-bit quantization error analysis for all the layers from the UNet of SD-v1.5.

respectively. For instance, for the SD-v1.5 UNet with 256 layers (excluding time embedding, the first and last layers), we get a total of 768 quantized candidates. We perform QAT over each candidate on a pre-defined training sub dataset, and validate the incurred quantization error of each candidate by comparing it against the full-precision model (more details in App. B).

**Measuring Quantization Errors.** To find the appropriate way to interpret the quantization error, we analyze four metrics: *Mean-Squared-Error (MSE)* that quantifies the pixel-level discrepancies between images (generations from floating and the quantized model in our case), *LPIPS* [89] that assesses human-like perceptual similarity judgments, *PSNR* [23] that measures image quality by comparing the maximum possible power of a signal with the power of a corrupted noise, and *CLIP score* [66] that evaluates the correlation between an image and its language description. After collecting the scores (examples in Fig. 2b and Fig. 2c, full metrics are listed in App. F), we further measure the consistency of them by calculating the Pearson correlation [8] for different metrics under the same bit widths (in Tab. 1), and different bit widths under the same metric (in Tab. 2). With these empirical results, we draw the following two main observations.

**Observation 1**: *MSE, PSNR, and LPIPS show strong correlation and they correlate well with the visual perception of image quality.*

Tab. 1 shows that MSE is highly correlated with PSNR and LPIPS under the same bit width. Additionally, we observe a similar trend of per-layer quantization error under different bit widths, as in Tab. 2. As for visual qualities in Fig. 2a and 2b, we can see that higher MSE errors lead to severe image quality degradation, *e.g.*, the highlighted RB *conv shortcut*. Therefore, the MSE metric effectively reflects quality degradations incurred by quantization, and it is unnecessary to incorporate PSNR and LPIPS further.

**Observation 2**: *After low-bit quantization, changes in CLIP score are not consistently correlated with MSE across different layers. Although some layers show smaller MSE, they may experience larger semantic degradation, reflected in larger CLIP score changes.*

We notice that, after quantization, the CLIP score changes for all layers only have a weak correlation with MSE, illustrated in Tab. 1. Some layers display smaller MSE but larger changes in CLIP score.

Table 1: Pearson correlation (absolute value) of quantization error between different metrics (*e.g.*, MSE *vs.* PSNR denotes the correlation between two metrics) when quantizing individual layers to 1, 2, and 3 bits. CS denotes CLIP Score.

| | MSE *vs.* PSNR | MSE *vs.* LPIPS | MSE *vs.* CS |
|---|---|---|---|
| 1 bit | 0.870 | 0.984 | 0.733 |
| 2 bit | 0.882 | 0.989 | 0.473 |
| 3 bit | 0.869 | 0.991 | 0.535 |

Table 2: Pearson correlation (absolute value) of quantization error between different bit pairs (*e.g.*, 1 *vs.* 2 denotes the correlation between the two bit widths) for a single metric when quantizing individual layers to 1, 2, and 3 bits.

| | MSE | PSNR | LPIPS | CLIP Score |
|---|---|---|---|---|
| 1 *vs.* 2 bit | 0.929 | 0.954 | 0.943 | 0.504 |
| 1 *vs.* 3 bit | 0.766 | 0.843 | 0.802 | 0.344 |
| 2 *vs.* 3 bit | 0.887 | 0.923 | 0.895 | 0.428 |

For example, in Fig. 2b, the MSE of CA *tok* layer ($5_{th}$ highlighted layer (green) from left to right) is less than that of RB *conv* layer ($6_{th}$ highlighted layer (orange) from left to right), yet the changes in CLIP score are the opposite. As observed in the first row of Fig. 2a, compared to RB *conv* layer, quantizing this CA *tok* layer changes the image content from "a teddy bear" to "a person", which diverges from the text prompt *A teddy bear on a skateboard in Times Square, doing tricks on a cardboard box ramp.* This occurs because MSE measures only the difference between two images, which does not capture the semantic degradation. In contrast, the CLIP score reflects the quantization error in terms of semantic information between the text and image. Thus, we employ the CLIP score as a complementary metric to represent the quantization error.

### 3.3 Deciding the Optimal Precision

With the above observations, we then develop the strategy for bit-width assignments. We select MSE and CLIP as our quantitative metrics, along with the number of parameters of each layer as the indicator of size savings.

**Assigning bits based on MSE.** Intuitively, layers with more parameters and lower quantization error are better candidates for extremely low-bit quantization, as the overall bit widths of the model can be significantly reduced. According to this, we propose a layer size-aware sensitivity score $\mathcal{S}$. For the $i_{th}$ layer, its sensitivity score for the $b$-bits ($b \in \{1, 2, 3\}$) is defined as $\mathcal{S}_{i,b} = M_{i,b}N_i^{-\eta}$, where $M$ denotes the MSE error, $N$ is the total number of parameters of the layer, and $\eta \in [0, 1]$ denotes the parameter size factor. To determine the bit width (*i.e.*, $b^*$) for each layer, we define a sensitivity threshold as $\mathcal{S}_o$, and the $i_{th}$ layer is assigned to $b_i^*$-bits, where $b_i^* = \min\{b|\mathcal{S}_{i,b} < \mathcal{S}_o\}$. The remaining layers are 4 bits.

**Assigning bits based on CLIP score.** For the layers with a high CLIP score dropping after quantization, instead of assigning bits based on sensitivity score as discussed above, we directly assign higher bits to those layers. Therefore, the quantized model can produce content that aligns with the semantic information of the prompt. We provide the detailed mixed-precision algorithm in Alg. 1 of App. B.

## 4 Training Extreme Low-bit Diffusion Model

With the bits of each layer decided, we then train the quantized model with a series of techniques to improve performance. The overview of our approach is illustrated in Fig. 3.

### 4.1 Initializing the Low-bit Diffusion Model

**Time Embedding Pre-computing and Caching.** During the inference time of a diffusion model, a time step $t$ is transformed into an embedding through projection layers to be incorporated into the diffusion model. As mentioned by existing works [27], the quantization of the projection layers can lead to large quantization errors. However, the embedding from each time step $t$ is always the same, suggesting that we can actually pre-compute the embedding *offline* and load cached values during inference, instead of computing the embedding every time. Furthermore, the storage size of the time embedding is $25.6\times$ smaller than the projection layers. Therefore, we pre-compute the time embedding and save the model without the project layers. More details are provided in App. C.

**Adding Balance Integer.** In general, weight distributions in deep neural networks are observed as symmetric around zero [94]. To validate the assumption on SD-v1.5, we analyze its weight

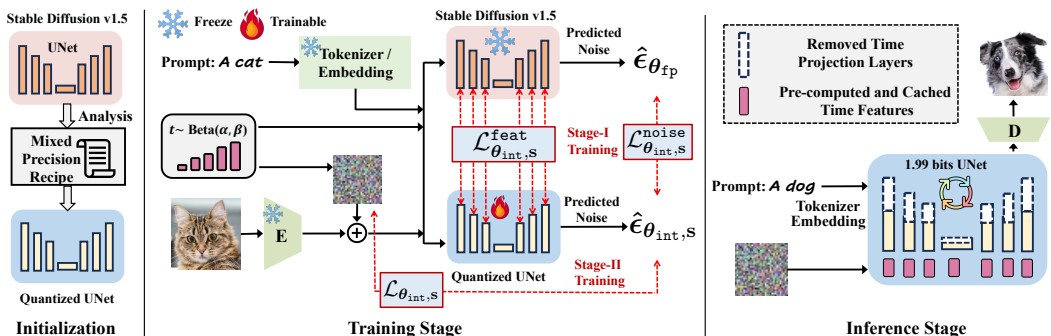

Figure 3: **Overview of the training and inference pipeline for the proposed BitsFusion.** *Left:* We analyze the quantization error for each layer in SD-v1.5 (Sec. 3.2) and derive the mixed-precision recipe (Sec. 3.3) to assign different bit widths to different layers. We then initialize the quantized UNet by adding a balance integer, pre-computing and caching the time embedding, and alternately optimizing the scaling factor (Sec. 4.1). *Middle:* During the Stage-I training, we freeze the teacher model (*i.e.*, SD-v1.5) and optimize the quantized UNet through CFG-aware quantization distillation and feature distillation losses, along with sampling time steps by considering quantization errors (Sec. 4.2). During the Stage-II training, we fine-tune the previous model with the noise prediction. *Right:* For the inference stage, using the pre-cached time features, our model processes text prompts and generates high-quality images.

distribution for the layers under full precision by calculating the skewness of weights. Notably, the skewness of more than $97\%$ of the layers ranges between $[-0.5, 0.5]$, indicating that the weight distributions are symmetric in almost all layers. Further details are provided in App. D.

However, existing works on diffusion model quantization overlook the symmetric property [38, 73, 39], as they perform relatively higher bits quantization, *e.g.*, $4$ or $8$ bits. This will hurt the model performance at extremely low bit levels. For example, in $1$-bit quantization, the possible most symmetric integer outcomes can only be $\{0, 1\}$ or $\{-1, 0\}$. Similarly, for $2$-bit quantization, the most balanced mapping integers can be either $\{-2, -1, 0, 1\}$ or $\{-1, 0, 1, 2\}$, significantly disrupting the symmetric property. The absence of a single value among $2$ or $4$ numbers under low-bit quantization can have a significant impact. To tackle this, we leverage the bit balance strategy [37, 56] to initialize the model. Specifically, we introduce an additional value to balance the original quantization values. Namely, in a $1$-bit model, we adjust the candidate integer set from $\{0, 1\}$ to $\{-1, 0, 1\}$, achieving a more balanced distribution. By doing so, we treat the balanced $n$-bits weights as $\texttt{log}(2^n + 1)$-bits.

**Scaling Factor Initialization via Alternating Optimization.** Initializing scaling factors is an important step in quantization. Existing QAT works typically employ the Min-Max initialization strategy [17, 52] to ensure the outliers are adequately represented and preserved. However, such a method faces challenges in extremely low-bit quantization settings like $1$-bit, since the distribution of the full-precision weights is overlooked, leading to a large quantization error and the increased difficulty to converge. Therefore, we aim to minimize the $\ell_2$ error between the quantized weights and full-precision weights with the optimization objective as:

$$\min_{\mathbf{s}} \|\mathbf{s} \cdot (\boldsymbol{\theta}_{\texttt{int}} - I_z) - \boldsymbol{\theta}_{\texttt{fp}}\|^2. \tag{4}$$

Nevertheless, considering the rounding operation, calculating an exact closed-form solution is not straightforward [29]. Inspired by the Lloyd-Max algorithm [28, 54], we use an optimization method on scaling factor $\mathbf{s}$ to minimize the initialization error of our quantized diffusion model as follows:

$$\boldsymbol{\theta}_{\texttt{int}}^j = Q_{\texttt{int}}(\boldsymbol{\theta}_{\texttt{fp}}, \mathbf{s}^{j-1}); \quad \mathbf{s}^j = \frac{\boldsymbol{\theta}_{\texttt{fp}}^j (\boldsymbol{\theta}_{\texttt{int}}^j - I_z)^\intercal}{(\boldsymbol{\theta}_{\texttt{int}}^j - I_z)(\boldsymbol{\theta}_{\texttt{int}}^j - I_z)^\intercal}, \tag{5}$$

where $Q_{\texttt{int}}(\cdot)$ denotes the integer mapping quantization operation that converts the full-precision weights to integer as Eq. (1), and $j$ represents the iterative step. The optimization is done for $10$ steps.

## 4.2 Two-Stage Training Pipeline

With the mixed-precision model initialized, we introduce the *two-stage* training pipeline. In Stage-I, we train the quantized model using the full-precision model as the teacher through distillation loss. In Stage-II, we fine-tune the model from the previous stage using noise prediction [21, 80].

**CFG-aware Quantization Distillation.** Similar to existing works [11], we fine-tune the quantized diffusion model to improve the performance. Here both the weights and scaling factors are optimized. Additionally, we notice that training the quantized model in a distillation fashion using the full-precision model yields better performance than training directly with vanilla noise prediction. Furthermore, during distillation, it is crucial for the quantized model to be aware of CFG, *i.e.*, text dropping is applied during distillation. Specifically, our training objective is as follows:

$$\mathcal{L}^{\texttt{noise}}_{\boldsymbol{\theta}_{\texttt{int},\mathbf{s}}} = \mathbb{E}_{t,\mathbf{x}} \left[ \| \hat{\boldsymbol{\epsilon}}_{\boldsymbol{\theta}_{\texttt{fp}}}(t, \mathbf{z}_t, \mathbf{c}) - \hat{\boldsymbol{\epsilon}}_{\boldsymbol{\theta}_{\texttt{int},\mathbf{s}}}(t, \mathbf{z}_t, \mathbf{c}) \| \right], \mathbf{c} = \varnothing \text{ if } P \sim U[0,1] < p \text{ else } \mathbf{c}, \quad (6)$$

where $P$ controls the text dropping probability during training and $p$ is set as 0.1.

**Feature Distillation.** To further improve the generation quality of the quantized model, we distill the full-precision model at a more fine-grained level through feature distillation [32] as follows:

$$\mathcal{L}^{\texttt{feat}}_{\boldsymbol{\theta}_{\texttt{int},\mathbf{s}}} = \mathbb{E}_{t,\mathbf{x}} \left[ \| \mathcal{F}_{\boldsymbol{\theta}_{\texttt{fp}}}(t, \mathbf{z}_t, \mathbf{c}) - \mathcal{F}_{\boldsymbol{\theta}_{\texttt{int},\mathbf{s}}}(t, \mathbf{z}_t, \mathbf{c}) \| \right], \quad (7)$$

where $\mathcal{F}_{\boldsymbol{\theta}}(\cdot)$ denotes the operation for getting features from the Down and Up blocks in UNet. We then have the overall distillation loss $\mathcal{L}^{\texttt{dist}}$ in Stage-I as follows:

$$\mathcal{L}^{\texttt{dist}} = \mathcal{L}^{\texttt{noise}}_{\boldsymbol{\theta}_{\texttt{int},\mathbf{s}}} + \lambda \mathcal{L}^{\texttt{feat}}_{\boldsymbol{\theta}_{\texttt{int},\mathbf{s}}}, \quad (8)$$

where $\lambda$ is empirically set as 0.01 to balance the magnitude of the two loss functions.

**Quantization Error-aware Time Step Sampling.** The training of diffusion models requires sampling different time steps in each optimization iteration. We explore how to adjust the strategy for time step sampling such that the quantization error in each time step can be effectively reduced during training. We first train a 1.99-bit quantized model with Eq. (8). Then, we calculate the difference of the predicted latent features between the quantized model and the full-precision model as $\mathbb{E}_{t,\mathbf{x}}[\frac{1-\bar{\alpha}_t}{\bar{\alpha}_t} \| \hat{\boldsymbol{\epsilon}}_{\boldsymbol{\theta}_{\texttt{fp}}}(t, \mathbf{z}_t, \mathbf{c}) - \hat{\boldsymbol{\epsilon}}_{\boldsymbol{\theta}_{\texttt{int},\mathbf{s}}}(t, \mathbf{z}_t, \mathbf{c}) \|^2]$, where $t \in [0, 1, \cdots, 999]$ and $\bar{\alpha}_t$ is the noise scheduler (detailed derivation in App. E). The evaluation is conducted on a dataset with 128 image-text pairs. Fig. 4 shows the quantization error does not distribute equally across all time steps. Notably, *the quantization error keeps increasing as the time steps approach* $t = 999$.

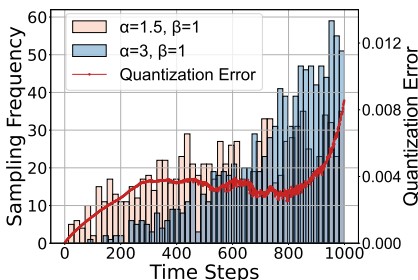

Figure 4: More time steps are sampled towards where larger quantization error occurs.

To mitigate the quantization error prevalent near the time steps $t = 999$, we propose a sampling strategy by utilizing a distribution specifically tailored to sample more time steps exhibiting the largest quantization errors, thereby enhancing performance. To achieve this goal, we leverage the Beta distribution. Specifically, time steps are sampled according to $t \sim Beta(\alpha, \beta)$, as shown in Fig. 4. We empirically set $\alpha = 3.0$ and $\beta = 1.0$ for the best performance. Combining the strategy of time steps sampling with Eq. (8), we conduct the Stage-I training.

**Fine-tuning with Noise Prediction.** After getting the model trained with the distillation loss in Stage-I, we then fine-tune it with noise prediction, as in Eq. (2), in Stage-II. We apply a text dropping with probability as 10% and modify the distribution of time step sampling based on the quantization error, as introduced above. The reason we leverage two-stage fine-tuning, instead of combining Stage-I and Stage-II, is that we observe more stabilized training results.

## 5 Experiments

**Implementation Details.** We develop our code using *diffusers* library[2] and train the models with AdamW optimizer [33] and a constant learning rate as $1e-05$ on an internal dataset. For Stage-I,

---

[2]https://github.com/huggingface/diffusers

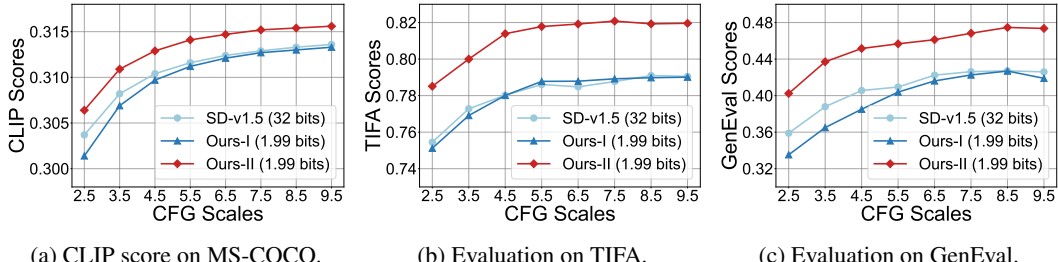

| (a) CLIP score on MS-COCO. | (b) Evaluation on TIFA. | (c) Evaluation on GenEval. |

Figure 5: Comparison between our 1.99-bits model *vs.* SD-v1.5 on various evaluation metrics with CFG scales ranging from 2.5 to 9.5. Ours-I denotes the model with Stage-I training and Ours-II denotes the model with Stage-II training.

Table 3: Comparison with existing quantization methods, including LSQ [11], Q-Diffusion [38], EfficientDM [17], and Apple-MBP [62]. The CLIP score is measured on 1K PartiPrompts.

| Method | Bit-width | CLIP score |
|---|---|---|
| SD-v1.5 | 32 | 0.3175 |
| LSQ | 2 | 0.2849 |
| Q-Diffusion | 4 | 0.3137 |
| EfficientDM | 2 | 0.2918 |
| Apple-MBP | 2 | 0.3023 |
| Ours | 1.99 | 0.3212 |

Table 4: Analysis of our proposed methods measured under various CFG scales, *i.e.*, 3.5, 5.5, 7.5, and 9.5. We use LSQ [11] as the basic QAT method, which involves the training of weights and scaling factors of a uniformly 2-bit quantized UNet. Then, we gradually introduce each proposed technique to evaluate their effectiveness. CLIP scores are measured on 1K PartiPrompts.

| Method | Bit-width | 3.5 | 5.5 | 7.5 | 9.5 | Average | Δ |
|---|---|---|---|---|---|---|---|
| SD-v1.5 | 32 | 0.3110 | 0.3159 | 0.3175 | 0.3180 | 0.3156 | - |
| QAT-Base | 2 | 0.2679 | 0.2793 | 0.2849 | 0.2868 | 0.2797 | - |
| +Balance | 2.32 | 0.2990 | 0.3059 | 0.3080 | 0.3086 | 0.3054 | +0.0257 |
| +Alternating Opt. | 2.32 | 0.3061 | 0.3108 | 0.3117 | 0.3115 | 0.3100 | +0.0046 |
| +Mixed/Caching | 1.99 | 0.3055 | 0.3129 | 0.3142 | 0.3145 | 0.3118 | +0.0018 |
| +Feat Dist. | 1.99 | 0.3086 | 0.3147 | 0.3167 | 0.3169 | 0.3142 | +0.0024 |
| +Time Sampling | 1.99 | 0.3098 | 0.3159 | 0.3181 | 0.3184 | 0.3156 | +0.0014 |
| +Fine-tuning | 1.99 | 0.3163 | 0.3192 | 0.3212 | 0.3205 | 0.3183 | +0.0027 |

we use 8 NVIDIA A100 GPUs with a total batch size of 256 to train the quantized model for 20K iterations. For Stage-II, we use 32 NVIDIA A100 GPUs with a total batch size of 1024 to train the quantized model for 50K iterations. During inference, we adopt the PNDM scheduler [49] with 50 sampling steps to generate images for comparison. Other sampling approaches (*e.g.*, DDIM [78] and DPMSolver [55]) lead to the same conclusion (App. K).

**Evaluation Metrics.** We conduct evaluation on CLIP Score and FID on MS-COCO [47], TIFA [26], GenEval [14], and human evaluation on PartiPrompts [86]. We adopt `ViT-B/32` model [10] in CLIP score and the `Mask2Former(Swin-S-8×2)` [5] in GenEval. App. I provides details for the metrics.

## 5.1 Main Results

**Comparison with SD-v1.5.** Our quantized 1.99-bits UNet consistently *outperforms* the full-precision model across all metrics.

- 30**K MS-COCO 2014 Validation Set.** For the CLIP score, as demonstrated in Fig. 5a, attributed to the proposed mixed-precision recipe with the introduced initialization techniques and advanced training schemes in Stage-I, our 1.99-bits UNet, with a storage size of 219MB, achieves performance comparable to the original SD-v1.5. Following Stage-II training, our model surpasses the performance of the original SD-v1.5. With CFG scales ranging from 2.5 to 9.5, our model yields $0.002 \sim 0.003$ higher CLIP scores.
- **TIFA.** As shown in Fig. 5b, our 1.99-bits model with Stage-I training performs comparably to the SD-v1.5. With the Stage-II training, our model achieves better metrics over the SD-v1.5.
- **GenEval.** We show the comparison results for GenEval in Fig. 5c (detailed comparisons of GenEval score are presented in Appn. L). Our model outperforms SD-v1.5 for all CFG scales.
- **Human Evaluation.** With the question: *Given a prompt, which image has better aesthetics and image-text alignment?* More users prefer the images generated by our quantized model over SD-v1.5, with the ratio as $54.4\%$. The results are shown in Fig. 6. We provide a detailed comparison in App. J.

**Comparison with Other Quantization Approaches.** Additionally, we conduct the experiments by comparing our approach with other works including LSQ [11], Q-Diffusion [38], EfficientDM [17],

Table 5: Analysis of $\eta$ in the mixed-precision strategy.

| $\eta$ | 0 | 0.1 | 0.2 | 0.3 | 0.4 | 0.5 |
|---|---|---|---|---|---|---|
| CLIP score | 0.3155 | 0.3173 | 0.3162 | 0.3181 | 0.3171 | 0.3168 |

Table 6: Anlysis of $\lambda$ in distillation loss.

| $\lambda$ | 1 | 0.1 | 0.01 |
|---|---|---|---|
| CLIP score | 0.3164 | 0.3159 | 0.3181 |

Table 7: Analysis of $\alpha$ in time step-aware sampling.

| $\alpha$ | 1.5 | 2.0 | 3.0 |
|---|---|---|---|
| CLIP score | 0.3169 | 0.3173 | 0.3181 |

and Apple-MBP [62], as shown in Tab. 3. Our model achieves a higher CLIP score compared with all other works and better performance than SD-v1.5.

## 5.2 Ablation Analysis

Here we perform extensive analysis for our proposed method. We mainly evaluate different experimental settings using the CLIP score measured on 1K PartiPrompts [86].

**Analysis of the Proposed Techniques.** We adopt the LSQ [11] as the basic QAT method to update the weights and scaling factors of a uniform 2-bit UNet with Min-Max initialization. Results are presented in Tab. 4 with the following details:

- **+Balance.** By adding a balance integer, a 2-bit model that typically represents 4 integer values can now represent 5 integers, becoming a 2.32-bit model by $\log(4+1)$ bits. The average CLIP score has significantly increased from 0.2797 to 0.3054.
- **+Alternating Opt.** By further utilizing the scaling factor initialization via alternating optimization, the average CLIP score of the 2.32-bit model increases to 0.3100.
- **+Mixed/Caching.** By leveraging time embedding pre-computing and caching, we minimize the storage requirements for time embedding and projection layers by only retaining the calculated features. This significantly reduces the averaged bits. Combined with our mixed-precision strategy, this approach reduces the average bits from 2.32 to 1.99 bits and can even improve the performance, *i.e.*, CLIP score improved from 0.3100 to 0.3118.
- **+Feat Dist.** By incorporating the feature distillation loss, *i.e.*, Eq. (7), the model can learn more fine-grained information from the teacher model, improving CLIP score from 0.3118 to 0.3142.
- **+Time Sampling.** By employing a quantization error-aware sampling strategy at various time steps, the model focuses more on the time step near $t = 999$. With this sampling strategy, our 1.99-bits model performs very closely to, or even outperforms, the original SD-v1.5.
- **+Fine-tuning.** By continuing with Stage-II training that incorporates noise prediction, our 1.99-bits model consistently outperforms the SD-v1.5 across various guidance scales, improving the CLIP score to 0.3183.

**Effect of $\eta$ in Mixed-Precision Strategy.** Tab. 5 illustrates the impact of the parameter size factor $\eta$ (as discussed in Sec. 3.3) in determining the optimal mixed precision strategy. We generate six different mixed precision recipes with different $\eta$ with 20K training iterations for comparisons. Initially, we explore the mixed precision strategy determined with and without the parameter size factor. Setting $\eta = 0$ results in $N^{-\eta} = 1$, indicating that the mixed precision is determined without considering the impact of parameter size. The results show that neglecting the parameter size significantly degrades performance. Further, we empirically choose $\eta = 0.3$ in our experiments after comparing different values of $\eta$.

**Effect of $\lambda$ of Distillation Loss.** Tab. 6 illustrates the impact of the balance factor $\lambda$ for loss functions in Eq. (8). We empirically choose $\lambda = 0.01$ in our experiments after comparing the performance.

**Effect of $\alpha$ in Time Step-aware Sampling Strategy.** Tab. 7 illustrates the impact of the $\alpha$ for different Beta sampling distribution. As analyzed in Sec. 4.2, the quantization error increases near $t = 999$. To increase sampling probability near this time step, Beta distribution requires $\alpha > 1$ with $\beta = 1$. A larger $\alpha$ enhances the sampling probability near $t = 999$. Compared to $\alpha = 1.5$ and $\alpha = 2.0$, $\alpha = 3.0$ concentrates more on later time steps and achieves the best performance. We choose $\alpha = 3.0$ in our experiments.

**Analysis for Different Schedulers.** One advantage of our training-based quantization approach is that our quantized model consistently outperforms SD-v1.5 for various sampling approaches. We conduct extensive evaluations on TIFA to show we achieve better performance than SD-v1.5 for using both DDIM [78] and DPMSolver [55] to perform the sampling. More details are shown in App. K.

**FID Results.** As stated in SDXL [63] and PickScore [35], FID may not honestly reflect the actual performance of the model in practice. FID measures the average distance between generated images

and reference real images, which is largely influenced by the training datasets. Also, FID does not capture the human preference which is the crucial metric for evaluating text-to-image synthesis. We present FID results evaluated on the 30K MS-COCO 2014 validation set in Fig. 7. Our Stage-I model has a similar FID as SD-v1.5. However, as training progresses, although our Stage-II model is preferred by users, its FID score is higher than both Stage-I and SD-v1.5.

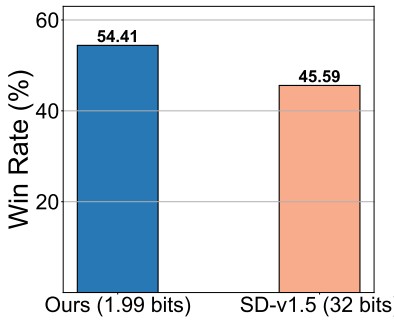

Figure 6: Overall human evaluation comparisons between SD-v1.5 and BitsFusion. Notably, BitsFusion, is favored 54.41% of the time over SD-v1.5.

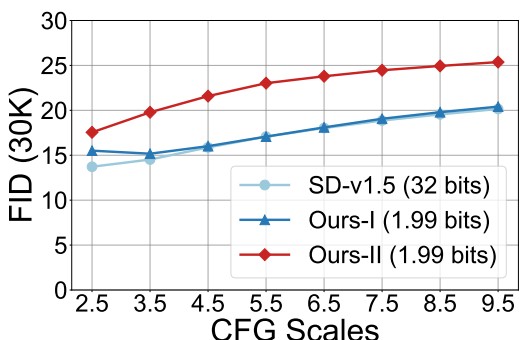

Figure 7: FID results evaluated on 30K MS-COCO 2014 validation set.

## 6   Conclusion

To enhance the storage efficiency of the large-scale diffusion models, we introduce an advanced weight quantization framework, BitsFusion, which effectively compresses the weights of UNet from SD-v1.5 to 1.99 bits, achieving a $7.9\times$ smaller model size. BitsFusion even outperforms SD-v1.5 in terms of generation quality. Specifically, we first conduct a comprehensive analysis to understand the impact of each layer during quantization and establish a mixed-precision strategy. Second, we propose a series of effective techniques to initialize the quantized model. Third, during the training stage, we enforce the quantized model to learn the full-precision SD-v1.5 by using distillation losses with the adjusted distribution of time step sampling. Finally, we fine-tune the previous quantized model through vanilla noise prediction. Our extensive evaluations on TIFA, GenEval, CLIP score, and human evaluation consistently demonstrate the advantage of BitsFusion over full-precision SD-v1.5.

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

# Appendix

## Table of Contents

## A Limitations

In this work, we study the storage size reduction of the UNet in Stable Diffusion v1.5 through weight quantization. The compression of VAE and CLIP text encoder [66] is also an interesting direction, which is not explored in this work. Additionally, our weight quantization techniques could be extended to the activations quantization, as a future exploration.

## B More details for Mixed-Precision Algorithm

In Sec. 3, we analyze the per-layer quantization error and develop the mixed-precision strategy. Here, we provide the detailed algorithm as outlined in Alg. 1. The inputs include: a pre-defined candidate set of bit-width $b \in \{1, 2, 3\}$, the full-precision SD-v1.5 $D$, the total number of layers $L$ (except for the time embedding, time projection, the first and last convolutional layers), the training dataset $X$, the number of training iterations $T$, the number of evaluation images for calculating metrics $K$, the bit threshold $\mathcal{S}_o$, the parameter size factor $\eta$, and the number of parameters of the $i_{th}$ layer $N_i$.

In the first stage, we aim to obtain quantized models by quantizing each individual layer. Given the full-precision SD-v1.5 UNet $D$, we consecutively perform the quantization on every single layer to 1, 2, or 3 bits individually, while maintaining the remaining layers at FP32 format. Notice, to align with our experiments, we add the balance integer and initialize the scaling factor with our alternating optimization. For each quantized model, the weights and scaling factors are fine-tuned using quantization-aware training to minimize the quantization error by learning the predicted noise of the SD-v1.5. We obtain quantized models $D_{i,b}, i = 1, 2, \cdots, L, b = 1, 2, 3$.

In the second stage, we measure the quantization error of each layer by calculating various metrics from comparing images generated by the quantized model $D_{i,b}$ with those from the unquantized SD-v1.5 $D$. Specifically, we generate $K = 100$ baseline images $I_d$ from the full-precision SD-v1.5 model with PartiPrompts. Then, for each quantized model $D_{i,b}$, we use identical prompts and seed to generate corresponding images $I_{i,b}$. We calculate the quantization error by measuring the metrics including MSE, CLIP score, PSNR, and LPIPS using these images and prompts.

In the third stage, we collect the mixed-precision recipe. We first compute a sensitivity score for each layer, factoring in both the MSE and the parameter size adjusted by $\eta$. For the $i_{th}$ layer, its sensitivity score for the $b$-bits ($b \in \{1, 2, 3\}$) is defined as $\mathcal{S}_{i,b} = M_{i,b} N_i^{-\eta}$, where $M$ denotes the MSE error, $N$ is the total number of parameters of the layer, and $\eta \in [0, 1]$ denotes the parameter size factor. To determine the bit width (i.e., $b^*$) for each layer, we define a sensitivity threshold as $\mathcal{S}_o$, and the $i_{th}$ layer is assigned to $b_i^*$-bits, where $b_i^* = \min\{b | \mathcal{S}_{i,b} < \mathcal{S}_o\}$. The remaining layers are set as 4 bits if they fail to meet the threshold. After determining the initial bits based on the MSE error, we refine this recipe by considering the degradation in the CLIP score associated with each bit-width. We simply consider the CLIP score change at 3 bits. We assign layers with the highest 10%, 5%, 2% CLIP score drop with 1, 2, 3 more bits, respectively.

The final output is a mixed-precision recipe $\{b_i^*\}, i = 1, 2, \cdots, L$, specifying the bit-width for each layer. Then, we set the first and last convolutional layers as 8 bits and pre-computing and caching the time embedding and projection layers.

## C More Details for Time Embedding Pre-computing and Caching

In Sec. 4.1, we introduce "Time Embedding Pre-computing and Caching". Here, we provide more details for the algorithm. In the Stable Diffusion model, the time step $t \in [0, 1, \cdots, 999]$ is transformed into a time embedding $\texttt{emb}_t$ through the equation $\texttt{emb}_t = e(t)$, where $e(t)$ denotes the time embedding layer and $\texttt{emb}_t \in \mathbb{R}^{d_{te}}$. In SD-v1.5, $d_{te} = 1280$. Then, for each $\texttt{ResBlock}$, denoted as $R_i$ for $i = 1, 2, \cdots, N_r$, where $N_r$ is total number of ResBlocks with time projection layers, the $\texttt{emb}_t$ is encoded by time projection layers $r_i(\cdot)$ by $\texttt{F}_{i,t} = r_i(\texttt{emb}_t)$. Notice that $r_i(\cdot)$ and $e(\cdot)$ are both linear layers. Finally, $\texttt{F}_{i,t}$ is applied to the intermediate activations of each $R_i$ via addition operation, effectively incorporating temporal information into the Stable Diffusion model.

As observed before [27], time embedding and projection layers exhibit considerable sensitivity to quantization during PTQ on DM. To address this problem, existing work specifically pays attention to reconstructing layers related to time embedding [27]. In this study, we propose a more effective

---

**Algorithm 1** Mixed-Precision Algorithm

---

**Input:** Candidate bits set $b \in \{1, 2, 3\}$, SD-v1.5 model $D$, number of total layers $L$ (except for the time embedding, time projection, the first and last convolutional layers), dataset $X$, training iterations $T$, number of evaluation images $K$, threshold $S_o$, parameter size factor $\eta$, number of parameters of the $i_{th}$ layer $N_i$.
**Output:** Mixed precision recipe $\{b_i^*\}, i = 1, 2, \cdots, L$.

 1: *1: Obtaining the quantized models.*
 2: **for** $b = 1$ to 3 **do**
 3:    **for** $i = 1$ to $L$ **do**
 4:       Quantize the $i$-th layer to $b$ bits via Eq. (1) and proposed initialization methods in Sec. 4.1 to get model $D_{i,b}$;
 5:       **for** $t = 1$ to $T$ **do**
 6:          Updating weights and scaling factors by minimizing the quantization error using quantization-aware training on $D_{i,b}$ with data $X$;
 7:       **end for**
 8:    **end for**
 9: **end for**
10: *2: Calculating quantization error metrics.*
11: Generating $K$ images $I_d$ via SD-v1.5;
12: **for** $b = 1$ to 3 **do**
13:    **for** $i = 1$ to $L$ **do**
14:       Generating $K$ images $I_{i,b}$ via quantized model $D_{i,b}$;
15:       Calculating MSE, $M_{i,b}$ via images $I_{i,b}$ and $I_d$;
16:       Calculating PSNR, $P_{i,b}$ via images $I_{i,b}$ and $I_d$;
17:       Calculating LPIPS, $L_{i,b}$ via images $I_{i,b}$ and $I_d$;
18:       Calculating CLIP score drop, $C_{i,b}$ via images $I_{i,b}$ and prompts;
19:    **end for**
20: **end for**
21: *2: Deciding the optimal precision.*
22: Calculating sensitivity score $\mathcal{S}_{i,b} = M_{i,b} N_i^{-\eta}$;
23: **for** $i = 1$ to $L$ **do**
24:    $b_i^* \leftarrow 4$;
25:    **for** $b = 3$ to 1 **do**
26:       **if** $\mathcal{S}_{i,b} < \mathcal{S}_o$ **then**
27:          Assign the $i$-th layer with $b$ bits with $b_i^* \leftarrow b$;
28:       **end if**
29:    **end for**
30: **end for**
31: Calculating CLIP score drop, $C_{i,3}$ and its $p_{th}$ percentile $C_p$;
32: **for** $i = 1$ to $L$ **do**
33:    **if** $C_{i,3} > C_{90}$ **then**
34:       $b_i^* \leftarrow b_i^* + 1$;
35:    **end if**
36:    **if** $C_{i,3} > C_{95}$ **then**
37:       $b_i^* \leftarrow b_i^* + 1$;
38:    **end if**
39:    **if** $C_{i,3} > C_{98}$ **then**
40:       $b_i^* \leftarrow b_i^* + 1$;
41:    **end if**
42: **end for**

---

method. We observe that 1) during the inference stage, for each time step $t$, the $\text{emb}_t$ and consequently $\text{F}_{i,t}$ remain constant. 2) In the Stable Diffusion model, the shape of $\text{F}_{i,t}$ are considerably smaller compared to time embedding and projection layers. Specifically, in SD-v1.5, $\text{F}_{i,t}$ is with the dimension in $\{320, 640, 1280\}$ which is largely smaller than time projection layers $W_r \in \mathbb{R}^{D \times 1280}$, where $D \in \{320, 640, 1280\}$. Therefore, we introduce an efficient and lossless method named Time Embedding Pre-computing and Caching. Specifically, for total $T_{\text{inf}}$ inference time steps, we opt to store only $T_{\text{inf}}$ time features, rather than retaining the original time embedding layers $e(\cdot)$ and the time projection layers in the $i$-th ResBlock $r_i(\cdot)$.

The inference time steps are set as 50 or less in most Stable Diffusion models. This method significantly reduces more than $1280/50 = 25.6\times$ storage requirements and entire computational costs in terms of time-related layers. Given that the storage size of the pre-computed $\text{F}_{i,t}$ is substantially

smaller than that of the original linear layers, this approach effectively diminishes the average bit of our quantized model without any performance degradation.

## D   Analysis of Symmetric Weight Distribution

In Sec. 4.1, we introduce "Adding Balance Integer" by assuming the weight distribution in Stable Diffusion is symmetric. Here, we provide more analysis for the assumption. To verify the weight distribution is symmetric around zero in SD-v1.5, we measure the skewness of the weight distribution of each layer. Lower skewness indicates a more symmetric weight distribution. As illustrated in Fig. 8, 97% of layers exhibiting skewness between [-0.5, 0.5], this suggests that most layers in SD-v1.5 have symmetric weight distributions.

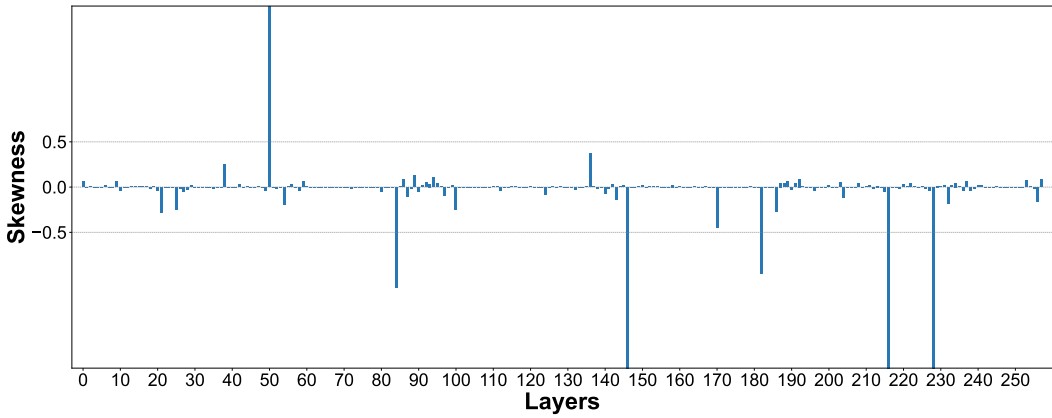

Figure 8: Skewness of weight distribution of each layer in SD-v1.5. Lower skewness represents the weight distribution is more symmetric. 97% layers are with skewness between [-0.5, 0.5], indicating that most layers have symmetric weight distribution in SD-v1.5.

## E   More Details for Quantization Error Across Different Time Steps

In Sec. 4.2, we introduce the "Quantization Error-aware Time Step Sampling" method. Here, we provide more details for measuring the quantization error from the predicted latent instead of the predicted noise. During the inference stage, the actual denoising step requires the scaling operation on the predicted noise in diffusion models. Therefore, directly calculating the quantization error via noise prediction is not accurate. Instead, we calculate the quantization error in the latent feature space. We derive the relationship of quantization error calculated from the predicted latent and noise as follows:

$$
\begin{aligned}
E &= \mathbb{E}_{t,\mathbf{x}} \left[ \left\| \hat{\mathbf{z}}_{\boldsymbol{\theta}_{\mathrm{fp}}}(t, \mathbf{z}_t, \mathbf{c}) - \hat{\mathbf{z}}_{\boldsymbol{\theta}_{\mathrm{int,s}}}(t, \mathbf{z}_t, \mathbf{c}) \right\|^2 \right], \\
&= \mathbb{E}_{t,\mathbf{x}} \left[ \left\| \left( \frac{1}{\sqrt{\bar{\alpha}_t}} \mathbf{z}_t - \frac{\sqrt{1 - \bar{\alpha}_t}}{\sqrt{\bar{\alpha}_t}} \hat{\boldsymbol{\epsilon}}_{\boldsymbol{\theta}_{\mathrm{fp}}}(t, \mathbf{z}_t, \mathbf{c}) \right) - \left( \frac{1}{\sqrt{\bar{\alpha}_t}} \mathbf{z}_t - \frac{\sqrt{1 - \bar{\alpha}_t}}{\sqrt{\bar{\alpha}_t}} \hat{\boldsymbol{\epsilon}}_{\boldsymbol{\theta}_{\mathrm{int,s}}}(t, \mathbf{z}_t, \mathbf{c}) \right) \right\|^2 \right], \\
&= \mathbb{E}_{t,\mathbf{x}} \left[ \frac{1 - \bar{\alpha}_t}{\bar{\alpha}_t} \left\| \hat{\boldsymbol{\epsilon}}_{\boldsymbol{\theta}_{\mathrm{fp}}}(t, \mathbf{z}_t, \mathbf{c}) - \hat{\boldsymbol{\epsilon}}_{\boldsymbol{\theta}_{\mathrm{int,s}}}(t, \mathbf{z}_t, \mathbf{c}) \right\|^2 \right],
\end{aligned}
\tag{9}
$$

where $\bar{\alpha}_t$ is the noise scheduler in [21].

# F   Detailed Metrics for Quantization Error by Quantizing Different Layers

In Sec. 3.2, we calculate the various metrics for representing the quantization error when quantizing different layers. Here, we provide detailed metrics when quantizing each layer of SD-v1.5 to 1, 2, and 3 bits.

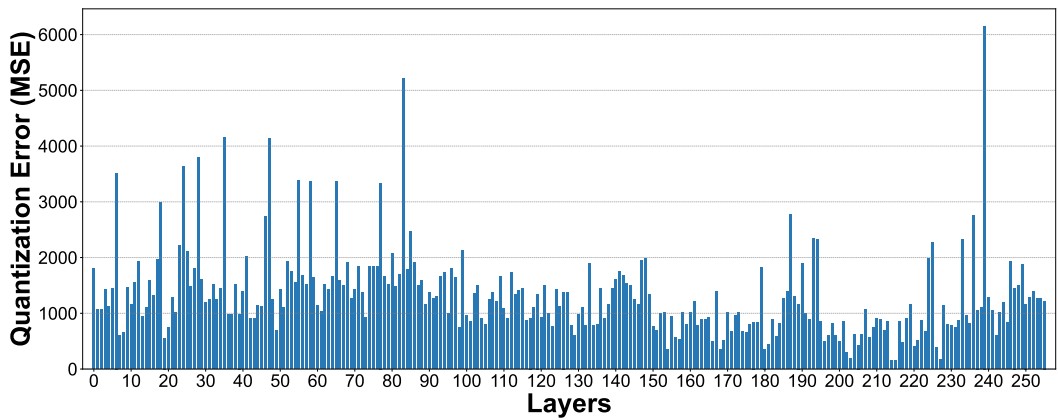

(a) MSE value caused by the 1-bit quantized layers in SD-v1.5.

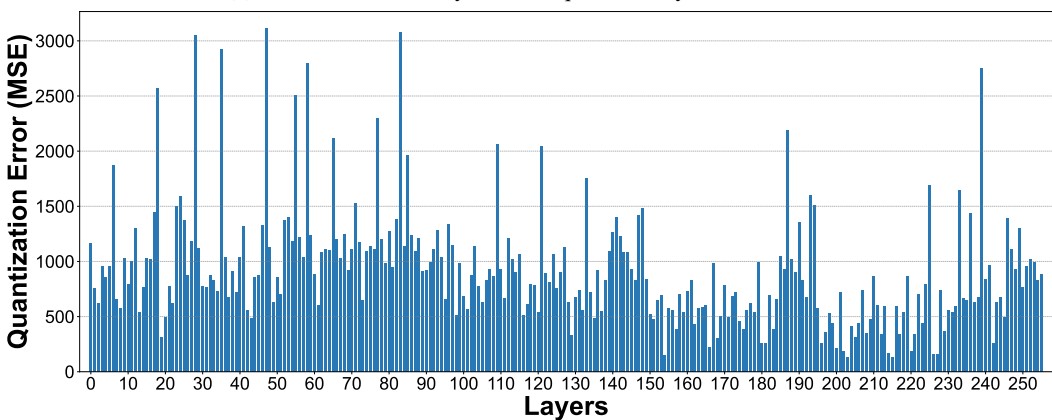

(b) MSE value caused by the 2-bit quantized layers in SD-v1.5.

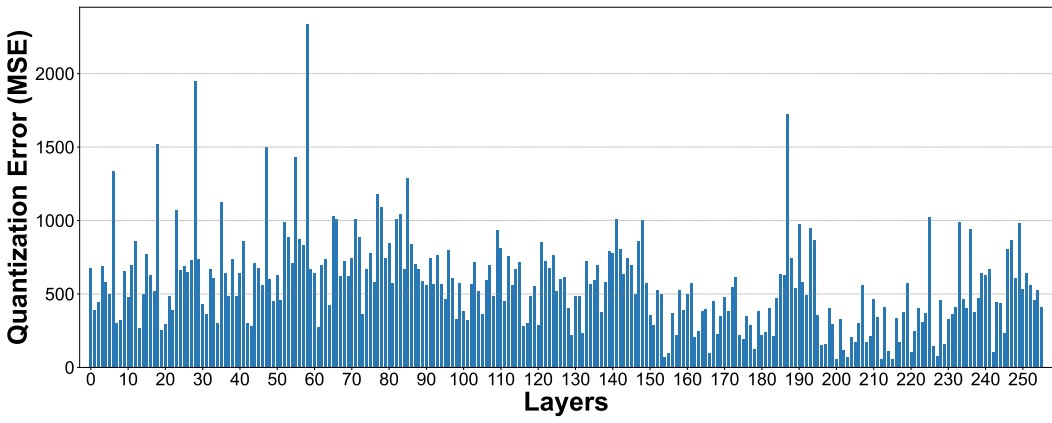

(c) MSE value caused by the 3-bit quantized layers in SD-v1.5.

Figure 9: MSE value caused by the quantized layers in SD-v1.5..

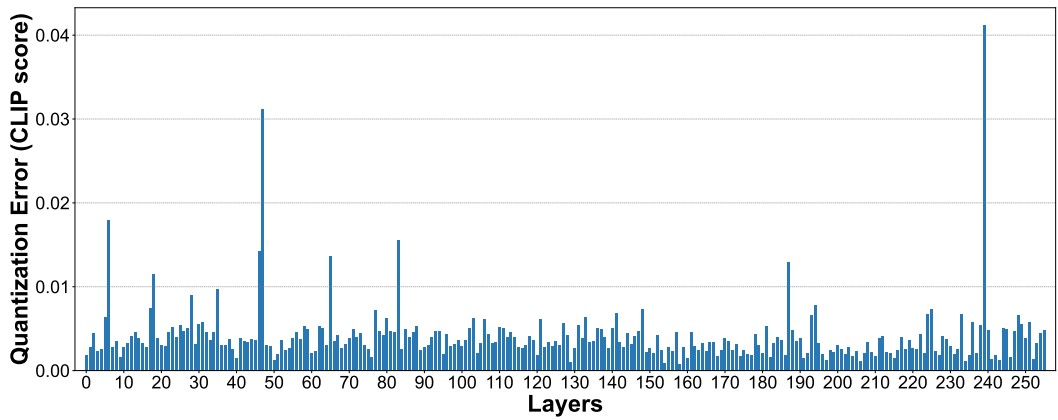

(a) CLIP score degradation caused by the 1-bit quantized layers in SD-v1.5.

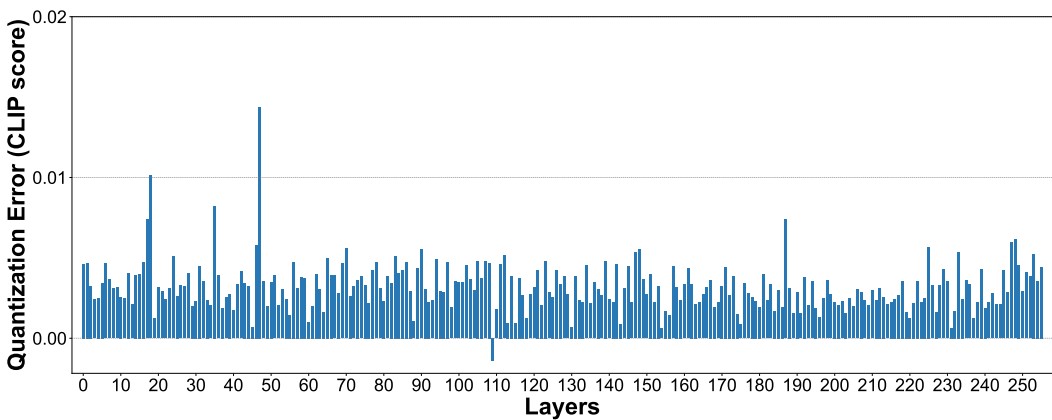

(b) CLIP score degradation caused by the 2-bit quantized layers in SD-v1.5.

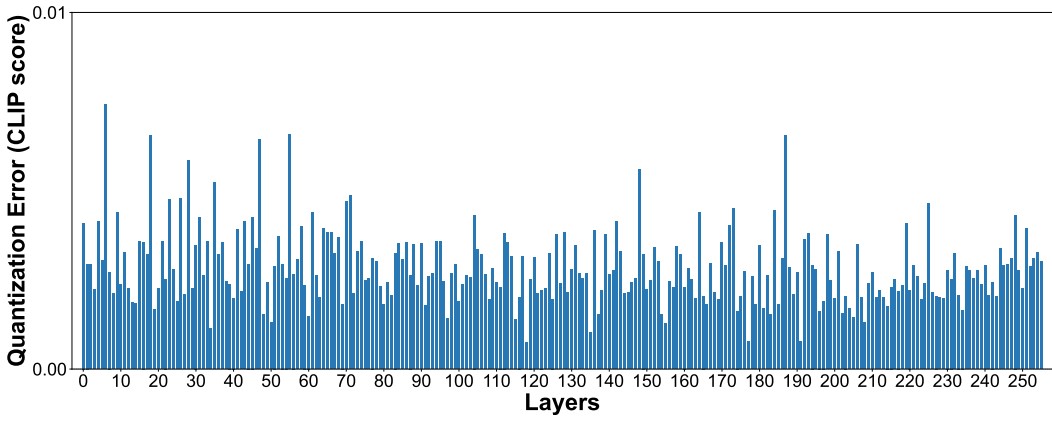

(c) CLIP score degradation caused by the 3-bit quantized layers in SD-v1.5.

Figure 10: CLIP score degradation caused by quantized layers in SD-v1.5.

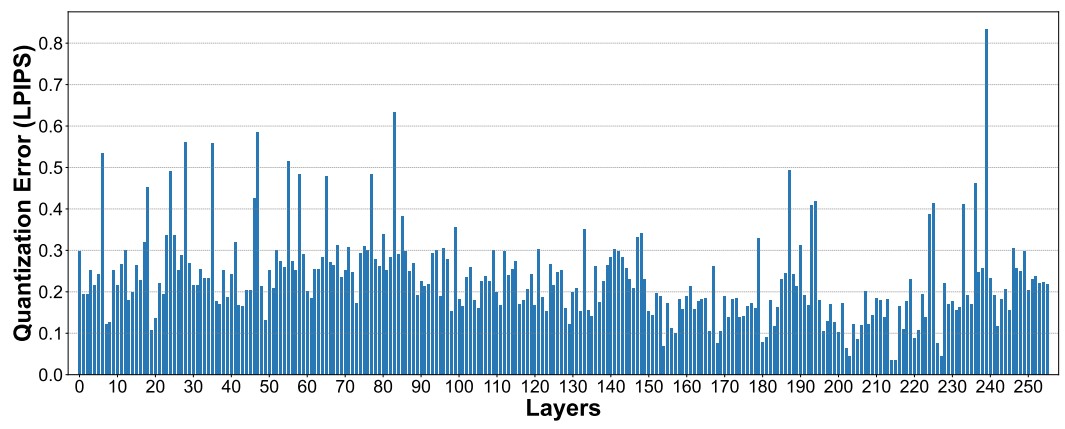

(a) LPIPS value of the 1-bit quantized layers in SD-v1.5.

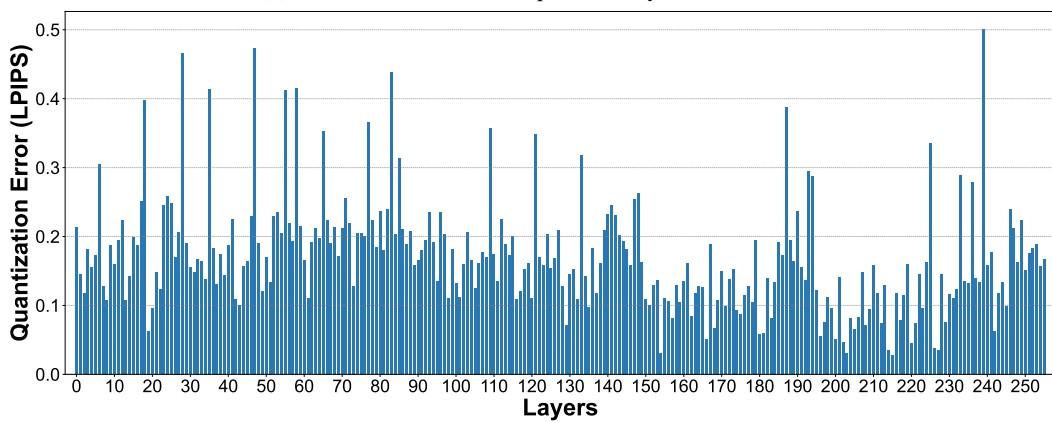

(b) LPIPS value of the 2-bit quantized layers in SD-v1.5.

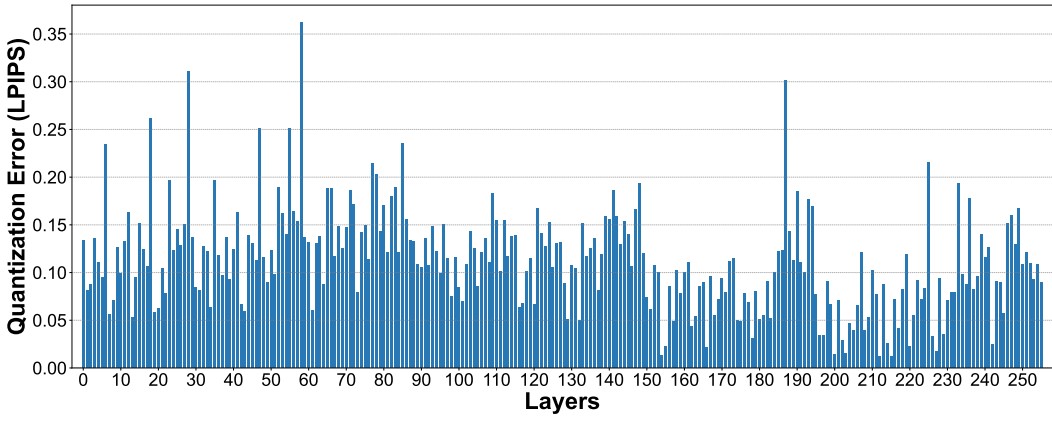

(c) LPIPS value of the 3-bit quantized layers in SD-v1.5.

Figure 11: LPIPS value of quantized layers in SD-v1.5.

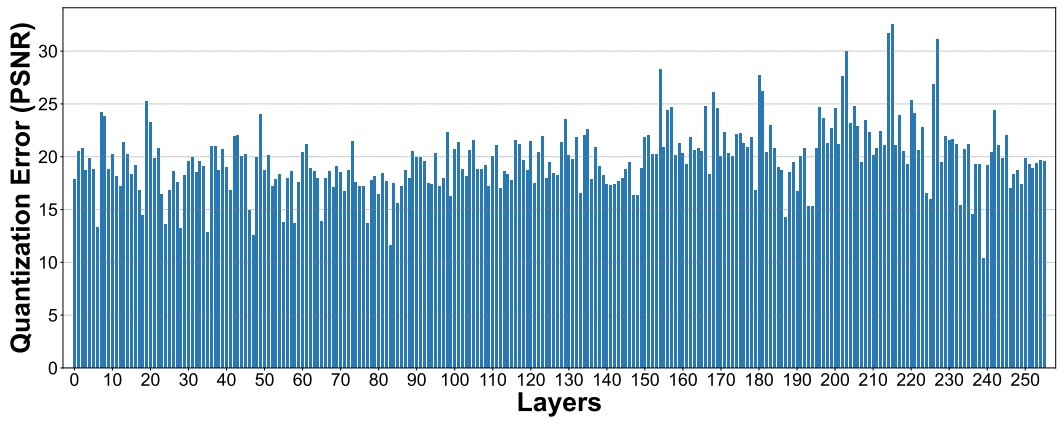

(a) PSNR value of the 1-bit quantized layers in SD-v1.5.

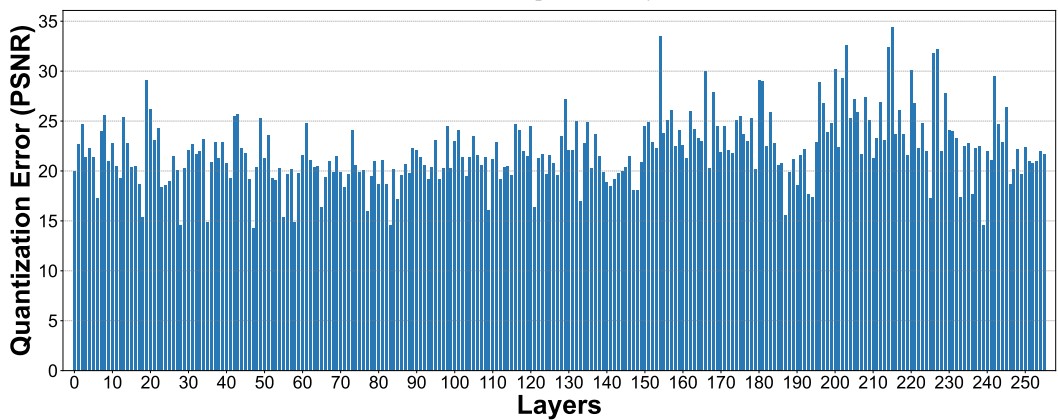

(b) PSNR value of the 2-bit quantized layers in SD-v1.5.

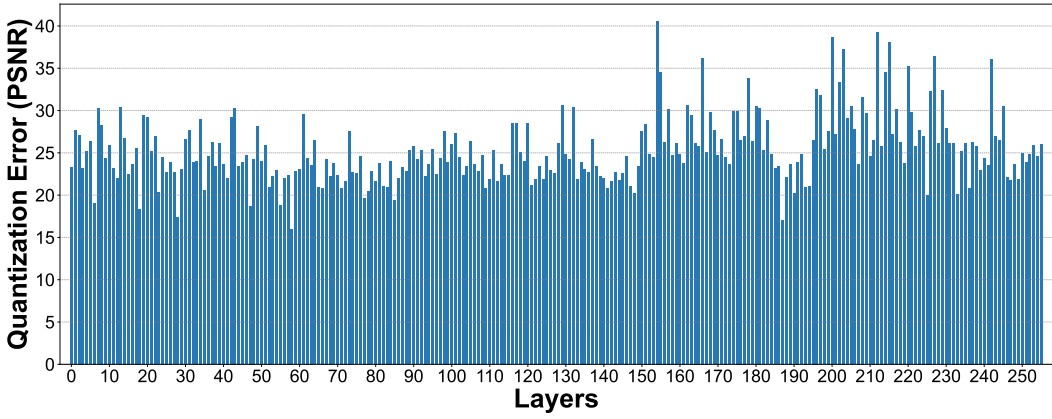

(c) PSNR value of the 3-bit quantized layers in SD-v1.5.

Figure 12: PSNR value of quantized layers in SD-v1.5.

# G   More Visualization for Quantization Error by Quantizing Different Layers

In Sec. 3.2, we show the images for representing the quantization error when quantizing different layers. Here, we provide more visualization for demonstrating the different quantization errors caused by quantizing different layers to 1 bit. The quantized layers from left to right correspond to the annotated layers at the bottom: SD-v1.5 w/o quantization, `Down.0.0.attn2.toq`, `Down.0.0.attn2.tok`, `Down.0.0.attn2.tov`, `Down.1.0.attn2.tok`, `Down.1.1.attn2.tok`, `Down.2.res.0.conv1`, `Up.2.res.2.convshortcut`, `Up.3.2.attn2.tok`, `Up.3.res.2.convshortcut`.

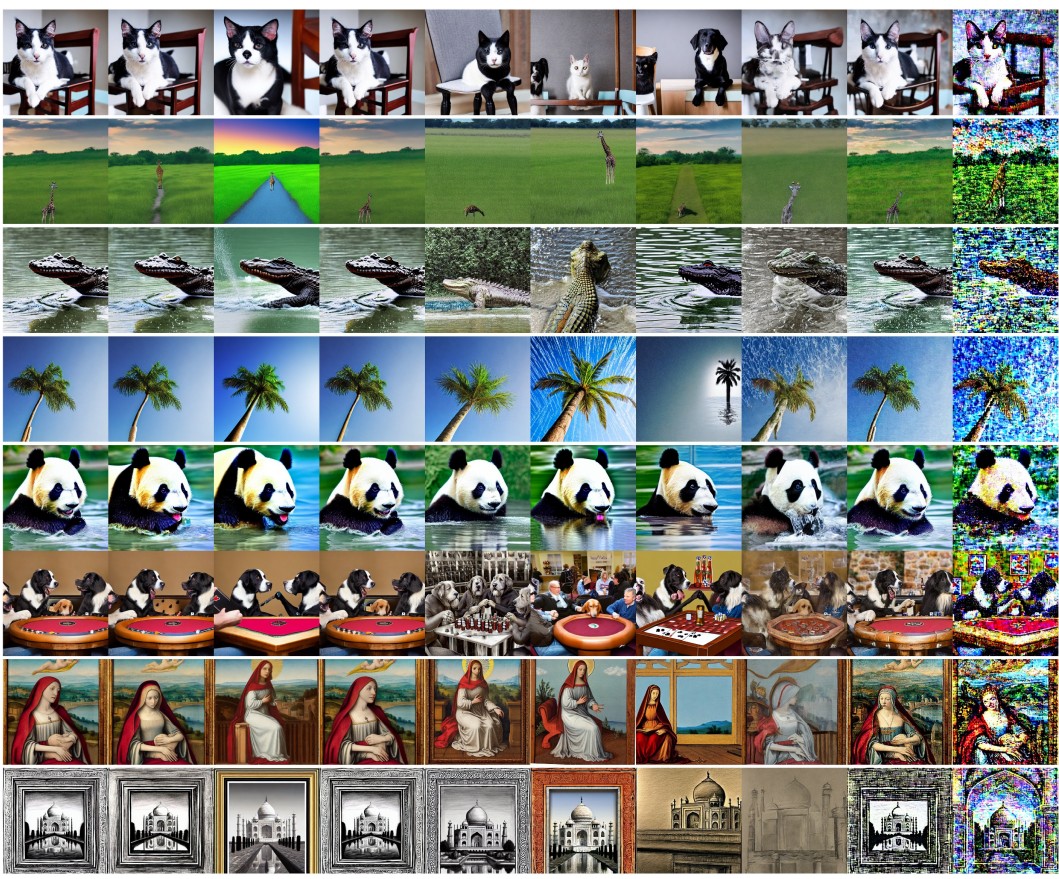

Figure 13: Quantization errors demonstrated in generated images (via PartiPrompts) after performing 1-bit quantization on different individual layers.

# H    1.99 Bits Mixed Precision Recipe

We provide our 1.99 bits recipe in our experiments. During the training and inference stage, we add a balancing integer to the $n$-bits values, resulting in $\log(2^n + 1)$ bits. We calculate the average bits by $\frac{\sum_i \log(2^{b_i^*}+1) \times N_i + 16 * N_{tf}}{N_w}$, where $b_i^*$ is the calculated bit-width in the $i_{th}$ layer, $N_i$ is the number of weights of the $i_{th}$ layer, $N_{tf}$ is the number of parameters for pre-cached time features, and $N_w$ is the total number of weights in linear and convolutional layers. We calculate the model size by integrating all other parameters as 32 bits. The index and name of each layer are listed:

```
1  down_blocks.0.attentions.0.proj_in: 6
2  down_blocks.0.attentions.0.transformer_blocks.0.attn1.to_q: 5
3  down_blocks.0.attentions.0.transformer_blocks.0.attn1.to_k: 5
4  down_blocks.0.attentions.0.transformer_blocks.0.attn1.to_v: 4
5  down_blocks.0.attentions.0.transformer_blocks.0.attn1.to_out.0: 6
6  down_blocks.0.attentions.0.transformer_blocks.0.attn2.to_q: 5
7  down_blocks.0.attentions.0.transformer_blocks.0.attn2.to_k: 7
8  down_blocks.0.attentions.0.transformer_blocks.0.attn2.to_v: 3
9  down_blocks.0.attentions.0.transformer_blocks.0.attn2.to_out.0: 3
10 down_blocks.0.attentions.0.transformer_blocks.0.ff.net.0.proj: 3
11 down_blocks.0.attentions.0.transformer_blocks.0.ff.net.2: 3
12 down_blocks.0.attentions.0.proj_out: 5
13 down_blocks.0.attentions.1.proj_in: 4
14 down_blocks.0.attentions.1.transformer_blocks.0.attn1.to_q: 3
15 down_blocks.0.attentions.1.transformer_blocks.0.attn1.to_k: 4
16 down_blocks.0.attentions.1.transformer_blocks.0.attn1.to_v: 6
17 down_blocks.0.attentions.1.transformer_blocks.0.attn1.to_out.0: 5
18 down_blocks.0.attentions.1.transformer_blocks.0.attn2.to_q: 5
19 down_blocks.0.attentions.1.transformer_blocks.0.attn2.to_k: 7
20 down_blocks.0.attentions.1.transformer_blocks.0.attn2.to_v: 2
21 down_blocks.0.attentions.1.transformer_blocks.0.attn2.to_out.0: 3
22 down_blocks.0.attentions.1.transformer_blocks.0.ff.net.0.proj: 3
23 down_blocks.0.attentions.1.transformer_blocks.0.ff.net.2: 3
24 down_blocks.0.attentions.1.proj_out: 6
25 down_blocks.0.resnets.0.conv1: 3
26 down_blocks.0.resnets.0.conv2: 3
27 down_blocks.0.resnets.1.conv1: 3
28 down_blocks.0.resnets.1.conv2: 4
29 down_blocks.0.downsamplers.0.conv: 4
30 down_blocks.1.attentions.0.proj_in: 4
31 down_blocks.1.attentions.0.transformer_blocks.0.attn1.to_q: 3
32 down_blocks.1.attentions.0.transformer_blocks.0.attn1.to_k: 3
33 down_blocks.1.attentions.0.transformer_blocks.0.attn1.to_v: 4
34 down_blocks.1.attentions.0.transformer_blocks.0.attn1.to_out.0: 4
35 down_blocks.1.attentions.0.transformer_blocks.0.attn2.to_q: 3
36 down_blocks.1.attentions.0.transformer_blocks.0.attn2.to_k: 5
37 down_blocks.1.attentions.0.transformer_blocks.0.attn2.to_v: 4
38 down_blocks.1.attentions.0.transformer_blocks.0.attn2.to_out.0: 3
39 down_blocks.1.attentions.0.transformer_blocks.0.ff.net.0.proj: 2
40 down_blocks.1.attentions.0.transformer_blocks.0.ff.net.2: 2
41 down_blocks.1.attentions.0.proj_out: 4
42 down_blocks.1.attentions.1.proj_in: 4
43 down_blocks.1.attentions.1.transformer_blocks.0.attn1.to_q: 2
44 down_blocks.1.attentions.1.transformer_blocks.0.attn1.to_k: 2
45 down_blocks.1.attentions.1.transformer_blocks.0.attn1.to_v: 4
46 down_blocks.1.attentions.1.transformer_blocks.0.attn1.to_out.0: 4
47 down_blocks.1.attentions.1.transformer_blocks.0.attn2.to_q: 3
48 down_blocks.1.attentions.1.transformer_blocks.0.attn2.to_k: 6
49 down_blocks.1.attentions.1.transformer_blocks.0.attn2.to_v: 4
50 down_blocks.1.attentions.1.transformer_blocks.0.attn2.to_out.0: 3
51 down_blocks.1.attentions.1.transformer_blocks.0.ff.net.0.proj: 2
52 down_blocks.1.attentions.1.transformer_blocks.0.ff.net.2: 2
53 down_blocks.1.attentions.1.proj_out: 4
54 down_blocks.1.resnets.0.conv1: 3
```

```
55 down_blocks.1.resnets.0.conv2: 3
56 down_blocks.1.resnets.0.conv_shortcut: 7
57 down_blocks.1.resnets.1.conv1: 3
58 down_blocks.1.resnets.1.conv2: 2
59 down_blocks.1.downsamplers.0.conv: 4
60 down_blocks.2.attentions.0.proj_in: 3
61 down_blocks.2.attentions.0.transformer_blocks.0.attn1.to_q: 3
62 down_blocks.2.attentions.0.transformer_blocks.0.attn1.to_k: 2
63 down_blocks.2.attentions.0.transformer_blocks.0.attn1.to_v: 3
64 down_blocks.2.attentions.0.transformer_blocks.0.attn1.to_out.0: 3
65 down_blocks.2.attentions.0.transformer_blocks.0.attn2.to_q: 3
66 down_blocks.2.attentions.0.transformer_blocks.0.attn2.to_k: 4
67 down_blocks.2.attentions.0.transformer_blocks.0.attn2.to_v: 4
68 down_blocks.2.attentions.0.transformer_blocks.0.attn2.to_out.0: 3
69 down_blocks.2.attentions.0.transformer_blocks.0.ff.net.0.proj: 2
70 down_blocks.2.attentions.0.transformer_blocks.0.ff.net.2: 1
71 down_blocks.2.attentions.0.proj_out: 3
72 down_blocks.2.attentions.1.proj_in: 4
73 down_blocks.2.attentions.1.transformer_blocks.0.attn1.to_q: 4
74 down_blocks.2.attentions.1.transformer_blocks.0.attn1.to_k: 2
75 down_blocks.2.attentions.1.transformer_blocks.0.attn1.to_v: 3
76 down_blocks.2.attentions.1.transformer_blocks.0.attn1.to_out.0: 3
77 down_blocks.2.attentions.1.transformer_blocks.0.attn2.to_q: 3
78 down_blocks.2.attentions.1.transformer_blocks.0.attn2.to_k: 4
79 down_blocks.2.attentions.1.transformer_blocks.0.attn2.to_v: 4
80 down_blocks.2.attentions.1.transformer_blocks.0.attn2.to_out.0: 3
81 down_blocks.2.attentions.1.transformer_blocks.0.ff.net.0.proj: 2
82 down_blocks.2.attentions.1.transformer_blocks.0.ff.net.2: 2
83 down_blocks.2.attentions.1.proj_out: 4
84 down_blocks.2.resnets.0.conv1: 3
85 down_blocks.2.resnets.0.conv2: 2
86 down_blocks.2.resnets.0.conv_shortcut: 4
87 down_blocks.2.resnets.1.conv1: 2
88 down_blocks.2.resnets.1.conv2: 1
89 down_blocks.2.downsamplers.0.conv: 1
90 down_blocks.3.resnets.0.conv1: 1
91 down_blocks.3.resnets.0.conv2: 1
92 down_blocks.3.resnets.1.conv1: 1
93 down_blocks.3.resnets.1.conv2: 1
94 up_blocks.0.resnets.0.conv1: 1
95 up_blocks.0.resnets.0.conv2: 2
96 up_blocks.0.resnets.0.conv_shortcut: 1
97 up_blocks.0.resnets.1.conv1: 1
98 up_blocks.0.resnets.1.conv2: 1
99 up_blocks.0.resnets.1.conv_shortcut: 1
100 up_blocks.0.resnets.2.conv1: 2
101 up_blocks.0.resnets.2.conv2: 1
102 up_blocks.0.resnets.2.conv_shortcut: 1
103 up_blocks.0.upsamplers.0.conv: 1
104 up_blocks.1.attentions.0.proj_in: 3
105 up_blocks.1.attentions.0.transformer_blocks.0.attn1.to_q: 2
106 up_blocks.1.attentions.0.transformer_blocks.0.attn1.to_k: 1
107 up_blocks.1.attentions.0.transformer_blocks.0.attn1.to_v: 2
108 up_blocks.1.attentions.0.transformer_blocks.0.attn1.to_out.0: 3
109 up_blocks.1.attentions.0.transformer_blocks.0.attn2.to_q: 3
110 up_blocks.1.attentions.0.transformer_blocks.0.attn2.to_k: 4
111 up_blocks.1.attentions.0.transformer_blocks.0.attn2.to_v: 4
112 up_blocks.1.attentions.0.transformer_blocks.0.attn2.to_out.0: 2
113 up_blocks.1.attentions.0.transformer_blocks.0.ff.net.0.proj: 2
114 up_blocks.1.attentions.0.transformer_blocks.0.ff.net.2: 2
115 up_blocks.1.attentions.0.proj_out: 3
116 up_blocks.1.attentions.1.proj_in: 3
117 up_blocks.1.attentions.1.transformer_blocks.0.attn1.to_q: 2
118 up_blocks.1.attentions.1.transformer_blocks.0.attn1.to_k: 2
119 up_blocks.1.attentions.1.transformer_blocks.0.attn1.to_v: 2
```

```
120 up_blocks.1.attentions.1.transformer_blocks.0.attn1.to_out.0: 2
121 up_blocks.1.attentions.1.transformer_blocks.0.attn2.to_q: 2
122 up_blocks.1.attentions.1.transformer_blocks.0.attn2.to_k: 4
123 up_blocks.1.attentions.1.transformer_blocks.0.attn2.to_v: 3
124 up_blocks.1.attentions.1.transformer_blocks.0.attn2.to_out.0: 1
125 up_blocks.1.attentions.1.transformer_blocks.0.ff.net.0.proj: 1
126 up_blocks.1.attentions.1.transformer_blocks.0.ff.net.2: 1
127 up_blocks.1.attentions.1.proj_out: 3
128 up_blocks.1.attentions.2.proj_in: 3
129 up_blocks.1.attentions.2.transformer_blocks.0.attn1.to_q: 1
130 up_blocks.1.attentions.2.transformer_blocks.0.attn1.to_k: 1
131 up_blocks.1.attentions.2.transformer_blocks.0.attn1.to_v: 2
132 up_blocks.1.attentions.2.transformer_blocks.0.attn1.to_out.0: 2
133 up_blocks.1.attentions.2.transformer_blocks.0.attn2.to_q: 1
134 up_blocks.1.attentions.2.transformer_blocks.0.attn2.to_k: 3
135 up_blocks.1.attentions.2.transformer_blocks.0.attn2.to_v: 2
136 up_blocks.1.attentions.2.transformer_blocks.0.attn2.to_out.0: 1
137 up_blocks.1.attentions.2.transformer_blocks.0.ff.net.0.proj: 1
138 up_blocks.1.attentions.2.transformer_blocks.0.ff.net.2: 1
139 up_blocks.1.attentions.2.proj_out: 2
140 up_blocks.1.resnets.0.conv1: 1
141 up_blocks.1.resnets.0.conv2: 1
142 up_blocks.1.resnets.0.conv_shortcut: 3
143 up_blocks.1.resnets.1.conv1: 1
144 up_blocks.1.resnets.1.conv2: 1
145 up_blocks.1.resnets.1.conv_shortcut: 3
146 up_blocks.1.resnets.2.conv1: 1
147 up_blocks.1.resnets.2.conv2: 1
148 up_blocks.1.resnets.2.conv_shortcut: 3
149 up_blocks.1.upsamplers.0.conv: 2
150 up_blocks.2.attentions.0.proj_in: 4
151 up_blocks.2.attentions.0.transformer_blocks.0.attn1.to_q: 2
152 up_blocks.2.attentions.0.transformer_blocks.0.attn1.to_k: 2
153 up_blocks.2.attentions.0.transformer_blocks.0.attn1.to_v: 3
154 up_blocks.2.attentions.0.transformer_blocks.0.attn1.to_out.0: 3
155 up_blocks.2.attentions.0.transformer_blocks.0.attn2.to_q: 1
156 up_blocks.2.attentions.0.transformer_blocks.0.attn2.to_k: 2
157 up_blocks.2.attentions.0.transformer_blocks.0.attn2.to_v: 1
158 up_blocks.2.attentions.0.transformer_blocks.0.attn2.to_out.0: 1
159 up_blocks.2.attentions.0.transformer_blocks.0.ff.net.0.proj: 1
160 up_blocks.2.attentions.0.transformer_blocks.0.ff.net.2: 1
161 up_blocks.2.attentions.0.proj_out: 3
162 up_blocks.2.attentions.1.proj_in: 4
163 up_blocks.2.attentions.1.transformer_blocks.0.attn1.to_q: 2
164 up_blocks.2.attentions.1.transformer_blocks.0.attn1.to_k: 3
165 up_blocks.2.attentions.1.transformer_blocks.0.attn1.to_v: 3
166 up_blocks.2.attentions.1.transformer_blocks.0.attn1.to_out.0: 3
167 up_blocks.2.attentions.1.transformer_blocks.0.attn2.to_q: 1
168 up_blocks.2.attentions.1.transformer_blocks.0.attn2.to_k: 3
169 up_blocks.2.attentions.1.transformer_blocks.0.attn2.to_v: 1
170 up_blocks.2.attentions.1.transformer_blocks.0.attn2.to_out.0: 1
171 up_blocks.2.attentions.1.transformer_blocks.0.ff.net.0.proj: 1
172 up_blocks.2.attentions.1.transformer_blocks.0.ff.net.2: 1
173 up_blocks.2.attentions.1.proj_out: 3
174 up_blocks.2.attentions.2.proj_in: 4
175 up_blocks.2.attentions.2.transformer_blocks.0.attn1.to_q: 2
176 up_blocks.2.attentions.2.transformer_blocks.0.attn1.to_k: 2
177 up_blocks.2.attentions.2.transformer_blocks.0.attn1.to_v: 2
178 up_blocks.2.attentions.2.transformer_blocks.0.attn1.to_out.0: 3
179 up_blocks.2.attentions.2.transformer_blocks.0.attn2.to_q: 2
180 up_blocks.2.attentions.2.transformer_blocks.0.attn2.to_k: 3
181 up_blocks.2.attentions.2.transformer_blocks.0.attn2.to_v: 1
182 up_blocks.2.attentions.2.transformer_blocks.0.attn2.to_out.0: 1
183 up_blocks.2.attentions.2.transformer_blocks.0.ff.net.0.proj: 1
184 up_blocks.2.attentions.2.transformer_blocks.0.ff.net.2: 1
```

```
185 up_blocks.2.attentions.2.proj_out: 3
186 up_blocks.2.resnets.0.conv1: 1
187 up_blocks.2.resnets.0.conv2: 2
188 up_blocks.2.resnets.0.conv_shortcut: 4
189 up_blocks.2.resnets.1.conv1: 1
190 up_blocks.2.resnets.1.conv2: 2
191 up_blocks.2.resnets.1.conv_shortcut: 4
192 up_blocks.2.resnets.2.conv1: 1
193 up_blocks.2.resnets.2.conv2: 1
194 up_blocks.2.resnets.2.conv_shortcut: 4
195 up_blocks.2.upsamplers.0.conv: 3
196 up_blocks.3.attentions.0.proj_in: 4
197 up_blocks.3.attentions.0.transformer_blocks.0.attn1.to_q: 2
198 up_blocks.3.attentions.0.transformer_blocks.0.attn1.to_k: 2
199 up_blocks.3.attentions.0.transformer_blocks.0.attn1.to_v: 6
200 up_blocks.3.attentions.0.transformer_blocks.0.attn1.to_out.0: 3
201 up_blocks.3.attentions.0.transformer_blocks.0.attn2.to_q: 2
202 up_blocks.3.attentions.0.transformer_blocks.0.attn2.to_k: 3
203 up_blocks.3.attentions.0.transformer_blocks.0.attn2.to_v: 1
204 up_blocks.3.attentions.0.transformer_blocks.0.attn2.to_out.0: 1
205 up_blocks.3.attentions.0.transformer_blocks.0.ff.net.0.proj: 1
206 up_blocks.3.attentions.0.transformer_blocks.0.ff.net.2: 1
207 up_blocks.3.attentions.0.proj_out: 4
208 up_blocks.3.attentions.1.proj_in: 4
209 up_blocks.3.attentions.1.transformer_blocks.0.attn1.to_q: 2
210 up_blocks.3.attentions.1.transformer_blocks.0.attn1.to_k: 3
211 up_blocks.3.attentions.1.transformer_blocks.0.attn1.to_v: 5
212 up_blocks.3.attentions.1.transformer_blocks.0.attn1.to_out.0: 3
213 up_blocks.3.attentions.1.transformer_blocks.0.attn2.to_q: 2
214 up_blocks.3.attentions.1.transformer_blocks.0.attn2.to_k: 3
215 up_blocks.3.attentions.1.transformer_blocks.0.attn2.to_v: 1
216 up_blocks.3.attentions.1.transformer_blocks.0.attn2.to_out.0: 1
217 up_blocks.3.attentions.1.transformer_blocks.0.ff.net.0.proj: 2
218 up_blocks.3.attentions.1.transformer_blocks.0.ff.net.2: 1
219 up_blocks.3.attentions.1.proj_out: 4
220 up_blocks.3.attentions.2.proj_in: 6
221 up_blocks.3.attentions.2.transformer_blocks.0.attn1.to_q: 2
222 up_blocks.3.attentions.2.transformer_blocks.0.attn1.to_k: 3
223 up_blocks.3.attentions.2.transformer_blocks.0.attn1.to_v: 4
224 up_blocks.3.attentions.2.transformer_blocks.0.attn1.to_out.0: 3
225 up_blocks.3.attentions.2.transformer_blocks.0.attn2.to_q: 4
226 up_blocks.3.attentions.2.transformer_blocks.0.attn2.to_k: 5
227 up_blocks.3.attentions.2.transformer_blocks.0.attn2.to_v: 1
228 up_blocks.3.attentions.2.transformer_blocks.0.attn2.to_out.0: 1
229 up_blocks.3.attentions.2.transformer_blocks.0.ff.net.0.proj: 3
230 up_blocks.3.attentions.2.transformer_blocks.0.ff.net.2: 2
231 up_blocks.3.attentions.2.proj_out: 4
232 up_blocks.3.resnets.0.conv1: 1
233 up_blocks.3.resnets.0.conv2: 2
234 up_blocks.3.resnets.0.conv_shortcut: 4
235 up_blocks.3.resnets.1.conv1: 2
236 up_blocks.3.resnets.1.conv2: 2
237 up_blocks.3.resnets.1.conv_shortcut: 4
238 up_blocks.3.resnets.2.conv1: 2
239 up_blocks.3.resnets.2.conv2: 2
240 up_blocks.3.resnets.2.conv_shortcut: 4
241 mid_block.attentions.0.proj_in: 2
242 mid_block.attentions.0.transformer_blocks.0.attn1.to_q: 3
243 mid_block.attentions.0.transformer_blocks.0.attn1.to_k: 1
244 mid_block.attentions.0.transformer_blocks.0.attn1.to_v: 2
245 mid_block.attentions.0.transformer_blocks.0.attn1.to_out.0: 2
246 mid_block.attentions.0.transformer_blocks.0.attn2.to_q: 1
247 mid_block.attentions.0.transformer_blocks.0.attn2.to_k: 4
248 mid_block.attentions.0.transformer_blocks.0.attn2.to_v: 4
249 mid_block.attentions.0.transformer_blocks.0.attn2.to_out.0: 3
```

```
250 mid_block.attentions.0.transformer_blocks.0.ff.net.0.proj: 2
251 mid_block.attentions.0.transformer_blocks.0.ff.net.2: 1
252 mid_block.attentions.0.proj_out: 3
253 mid_block.resnets.0.conv1: 1
254 mid_block.resnets.0.conv2: 1
255 mid_block.resnets.1.conv1: 1
256 mid_block.resnets.1.conv2: 1
conv_in: 8
conv_out: 8
```

# I  Details for Evaluation Metrics

In Sec. 5, we measure the performance on various metrics such as TIFA, GenEval, CLIP score and FID. Here, we provide more details for these metrics.

**TIFA Score.** TIFA v1.0 [26] aims to measure the faithfulness of generated images. It includes various 4K text prompts sampled from the MS-COCO captions [47], DrawBench [72], PartiPrompts [86], and PaintSkill [6], associated with a pre-generated set of question-answer pairs, resulting in 25K questions covering 4.5K diverse elements. Image faithfulness is measured by determining if the VQA model can accurately answer the questions from the generated images.

**GenEval Score.** GenEval [14] measures the consistency between the generated images and the description, including 6 different tasks: *single object*, *two object*, *counting*, *colors*, *position*, *color attribution*. All text prompts are generated from task-specific templates filled in with: randomly sampled object names from MS-COCO [47], colors from Berlin-Kay basic color theory, numbers with 2, 3, 4, and relative positions from "above", "below", "to the left of", or "to the right of". We adopt the pre-trained object detection model Mask2Former (Swin-S-8×2) [5] for evaluation.

**CLIP score and FID.** CLIP score measures measure the similarity between text prompts and corresponding generated images. FID is used to evaluate the quality of generated images by measuring the distance between the distributions of features extracted from generated images and target images. In the main experiments, evaluation are measured based on MS-COCO 2014 validation set with 30K image-caption pairs [47]. We adopt ViT-B/32 model to evaluate the CLIP score in our experiments.

# J  Human Evaluation

In Sec. 5, we provide the human evaluation results. Here, we provide more detailed human evaluation with category and challenge comparisons on PartiPrompts (P2), comparing Stable Diffusion v1.5 and BitsFusion, with the question: *Given a prompt, which image has better aesthetics and image-text alignment?* Our model is selected 888 times out of 1632 comparisons, indicating a general preference over SD-v1.5, which is chosen 744 times, demonstrating more appealing and accurate generated images.

## J.1  Analysis on Categories

**Illustrations, People, and Arts.** Our model significantly outperforms SD-v1.5 in generating illustrations (77 wins out of 124), images of people (101 out of 174), and arts (45 out of 65).

**Outdoor and Indoor Scenes.** Our model also shows strength in generating both outdoor (73 out of 131) and indoor scenes (23 out of 40), suggesting better environmental rendering capabilities.

## J.2  Analysis on Challenges

**Complex and Fine-grained Detail**: Our model excels in generating images with complex details (73 out of 113) and fine-grained details (173 out of 312), suggesting advanced capabilities in maintaining detail at varying complexity levels.

**Imagination and Style & Format**: Our model also shows a strong performance in tasks requiring imaginative (92 out of 149) and stylistic diversity (118 out of 204), highlighting its flexibility and creative handling of artistic elements.

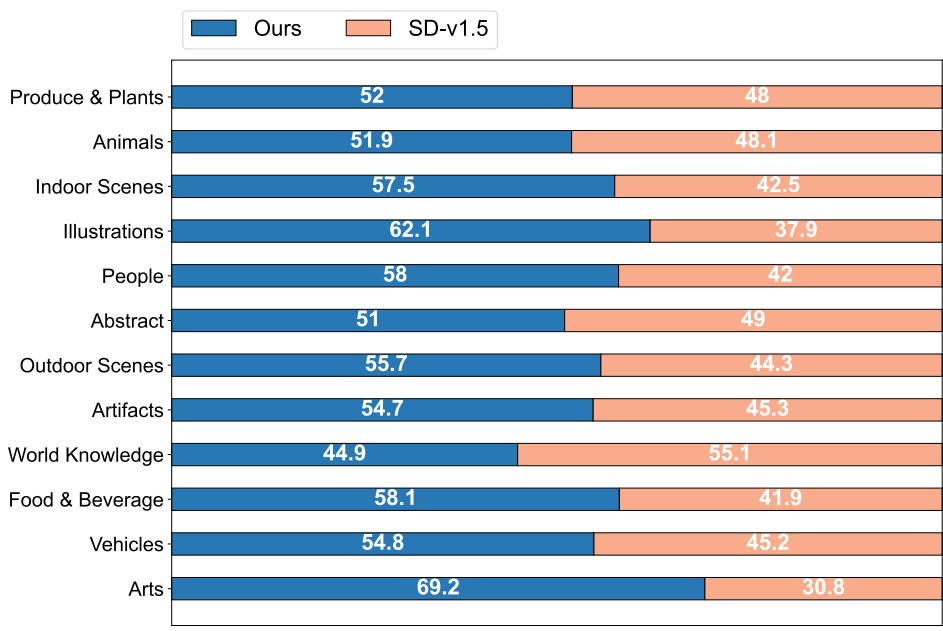

Figure 14: Human evaluation across particular categories.

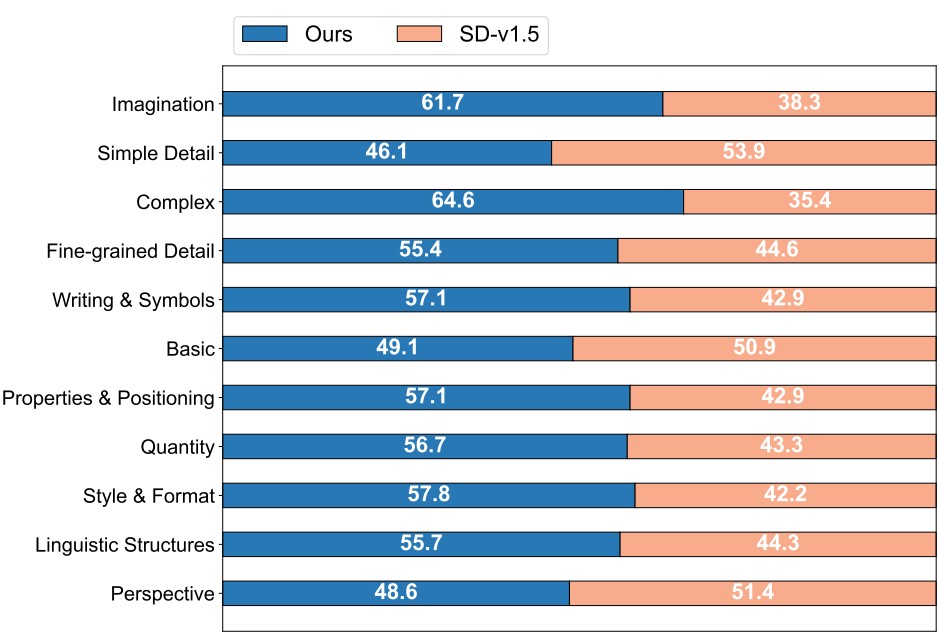

Figure 15: Human evaluation across particular challenges.

The strong performance in imaginative and artistic categories presents an opportunity to target applications in creative industries, such as digital art and entertainment, where these capabilities can be particularly valuable.

## K Evaluation on Different Schedulers

In the main experiments in Sec. 5, we leverage the PNDM scheduler to generate images. Here, we measured the performance of different schedulers, such as DDIM [78] and DPMSolver [55], to demonstrate the generality and effectiveness of BitsFusion. We set 50 inference steps and fix

the random seed as 1024. As shown in Fig. 16, BitsFusionconsistently outperforms SD-v1.5 with different schedulers.

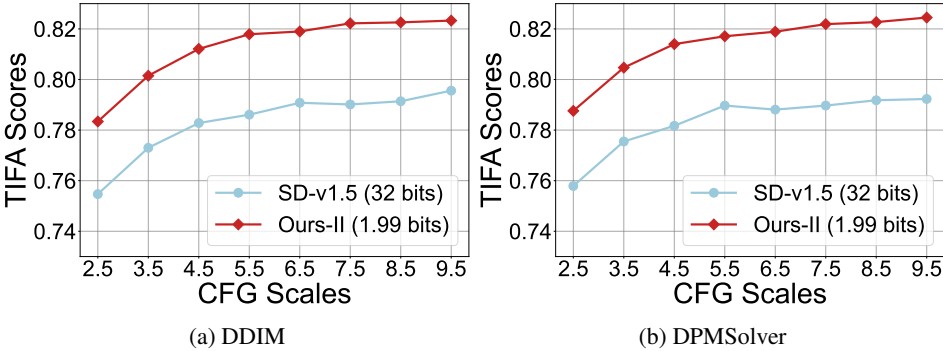

|  | (a) DDIM | (b) DPMSolver |
|---|---|---|

Figure 16: TIFA scores comparisons between SD-v1.5 and BitsFusion, with different schedulers. Left: TIFA scores measured with DDIM [78] scheduler. Right: TIFA score measured with DPMSolver [55] scheduler.

## L  Detailed GenEval Results

In Sec. 5, we provide the overall GenEval results. Here, we provide detailed GenEval results for further comparisons as illustrated in Tab. 8.

Table 8: Detailed GenEval with different CFG scales.

| Method | Overall | Single | Two | Counting | Colors | Position | Color Attribution |
|---|---|---|---|---|---|---|---|
| | | | Guidance Scale = 2.5 | | | | |
| SD-v1.5 | 0.3589 | 0.9350 | 0.2626 | 0.2775 | 0.6043 | 0.0340 | 0.0400 |
| Ours-I | 0.3353 | 0.9075 | 0.2444 | 0.2550 | 0.5426 | 0.0280 | 0.0340 |
| Ours-II | 0.4024 | 0.8975 | 0.3859 | 0.2750 | 0.6979 | 0.0560 | 0.1020 |
| | | | Guidance Scale = 3.5 | | | | |
| SD-v1.5 | 0.3879 | 0.9400 | 0.3010 | 0.3275 | 0.6787 | 0.0300 | 0.0500 |
| Ours-I | 0.3650 | 0.9500 | 0.2808 | 0.2575 | 0.6277 | 0.0280 | 0.0460 |
| Ours-II | 0.4370 | 0.9350 | 0.4727 | 0.3125 | 0.7340 | 0.0600 | 0.1080 |
| | | | Guidance Scale = 4.5 | | | | |
| SD-v1.5 | 0.4056 | 0.9700 | 0.3010 | 0.3200 | 0.7426 | 0.0340 | 0.0660 |
| Ours-I | 0.3851 | 0.9500 | 0.3091 | 0.3100 | 0.6574 | 0.0340 | 0.0500 |
| Ours-II | 0.4516 | 0.9575 | 0.4788 | 0.3450 | 0.7723 | 0.0520 | 0.1040 |
| | | | Guidance Scale = 5.5 | | | | |
| SD-v1.5 | 0.4094 | 0.9750 | 0.3111 | 0.3325 | 0.7319 | 0.0400 | 0.0660 |
| Ours-I | 0.4039 | 0.9675 | 0.3232 | 0.3425 | 0.7000 | 0.0300 | 0.0600 |
| Ours-II | 0.4567 | 0.9600 | 0.4909 | 0.3175 | 0.7979 | 0.0540 | 0.1200 |
| | | | Guidance Scale = 6.5 | | | | |
| SD-v1.5 | 0.4224 | 0.9800 | 0.3293 | 0.3725 | 0.7447 | 0.0400 | 0.0680 |
| Ours-I | 0.4161 | 0.9675 | 0.3414 | 0.3350 | 0.7425 | 0.0360 | 0.0740 |
| Ours-II | 0.4612 | 0.9750 | 0.4990 | 0.3275 | 0.7957 | 0.0540 | 0.1160 |
| | | | Guidance Scale = 7.5 | | | | |
| SD-v1.5 | 0.4262 | 0.9775 | 0.3313 | 0.3850 | 0.7596 | 0.0440 | 0.0600 |
| Ours-I | 0.4226 | 0.9775 | 0.3495 | 0.3600 | 0.7447 | 0.0360 | 0.0680 |
| Ours-II | 0.4682 | 0.9800 | 0.5091 | 0.3300 | 0.8085 | 0.0680 | 0.1140 |
| | | | Guidance Scale = 8.5 | | | | |
| SD-v1.5 | 0.4271 | 0.9825 | 0.3273 | 0.3925 | 0.7745 | 0.0320 | 0.0540 |
| Ours-I | 0.4269 | 0.9800 | 0.3616 | 0.3475 | 0.7702 | 0.0400 | 0.0620 |
| Ours-II | 0.4747 | 0.9700 | 0.5111 | 0.3675 | 0.8213 | 0.0620 | 0.1160 |
| | | | Guidance Scale = 9.5 | | | | |
| SD-v1.5 | 0.4260 | 0.9825 | 0.3556 | 0.3825 | 0.7553 | 0.0280 | 0.0520 |
| Ours-I | 0.4190 | 0.9825 | 0.3495 | 0.3450 | 0.7447 | 0.0300 | 0.0620 |
| Ours-II | 0.4736 | 0.9700 | 0.5192 | 0.3625 | 0.8277 | 0.0560 | 0.1060 |

# M    More Comparisons

We provide the prompts for the images featured in the Fig. 1. Additionally, we provide more generated images for the comparison.

## M.1    Prompts

Prompts of Fig. 1 from left to right are:

```
1. a portrait of an anthropomorphic cyberpunk raccoon smoking a cigar, cyberpunk!,
fantasy, elegant, digital painting, artstation, concept art, matte, sharp focus,
illustration, art by josan Gonzalez

2. Pirate ship trapped in a cosmic maelstrom nebula, rendered in cosmic beach
whirlpool engine, volumetric lighting, spectacular, ambient lights,
light pollution, cinematic atmosphere, art nouveau style,
illustration art artwork by SenseiJaye, intricate detail.

3. tropical island, 8 k, high resolution, detailed charcoal drawing,
beautiful hd, art nouveau, concept art, colourful, in the style of vadym meller

4. anthropomorphic art of a fox wearing a white suit, white cowboy hat,
and sunglasses, smoking a cigar, texas inspired clothing by artgerm,
victo ngai, ryohei hase, artstation. highly detailed digital painting,
smooth, global illumination, fantasy art by greg rutkowsky, karl spitzweg

5. a painting of a lantina elder woman by Leonardo da Vinci . details, smooth,
sharp focus, illustration, realistic, cinematic, artstation, award winning, rgb ,
unreal engine, octane render, cinematic light, macro, depth of field, blur,
red light and clouds from the back, highly detailed epic cinematic concept art CG
render made in Maya, Blender and Photoshop, octane render, excellent composition,
dynamic dramatic cinematic lighting, aesthetic, very inspirational, arthouse.

6. panda mad scientist mixing sparkling chemicals, high-contrast painting

7. An astronaut riding a horse on the moon, oil painting by Van Gogh.

8. A red dragon dressed in a tuxedo and playing chess. The chess pieces are
fashioned after robots.
```

## M.2    Additional Image Comparisons

We provide more images for further comparisons. For each set of two rows, the top row displays images generated using the full-precision Stable Diffusion v1.5, while the bottom row features images generated from BitsFusion, where the weights of UNet are quantized into 1.99 bits and the model size is $7.9\times$ smaller than the one from SD-v1.5. All the images are synthesized under the setting of using PNDM sampler with $50$ sampling steps and random seed as $1024$.

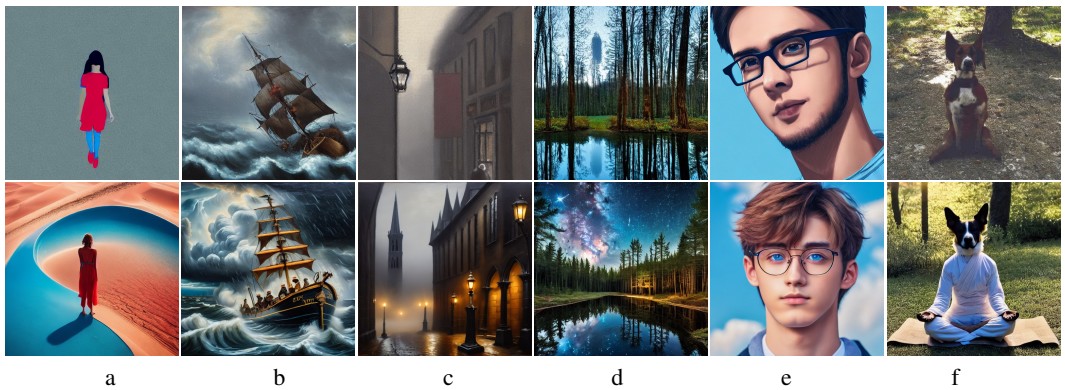

Figure 17: Top: Images generated from full-precision Stable Diffusion v1.5. Bottom: Images generated from BitsFusion. Prompts from left to right are: **a**: *A person standing on the desert, desert waves, gossip illustration, half red, half blue, abstract image of sand, clear style, trendy illustration, outdoor, top view, clear style, precision art, ultra high definition image*; **b**: *A detailed oil painting of an old sea captain, steering his ship through a storm. Saltwater is splashing against his weathered face, determination in his eyes. Twirling malevolent clouds are seen above and stern waves threaten to submerge the ship while seagulls dive and twirl through the chaotic landscape. Thunder and lights embark in the distance, illuminating the scene with an eerie green glow.*; **c**: *A solitary figure shrouded in mists peers up from the cobble stone street at the imposing and dark gothic buildings surrounding it. an old-fashioned lamp shines nearby. oil painting.*; **d**: *A deep forest clearing with a mirrored pond reflecting a galaxy-filled night sky*; **e**: *a handsome 24 years old boy in the middle with sky color background wearing eye glasses, it's super detailed with anime style, it's a portrait with delicated eyes and nice looking face*; **f**: *A dog that has been meditating all the time.*

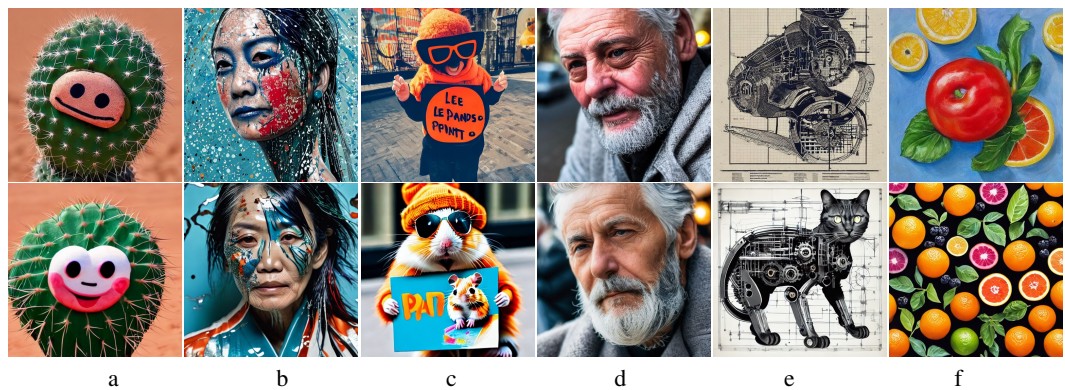

Figure 18: Top: Images generated from full-precision Stable Diffusion v1.5. Bottom: Images generated from BitsFusion. Prompts from left to right are: **a**: *A small cactus with a happy face in the Sahara desert.*; **b**: *A middle-aged woman of Asian descent, her dark hair streaked with silver, appears fractured and splintered, intricately embedded within a sea of broken porcelain. The porcelain glistens with splatter paint patterns in a harmonious blend of glossy and matte blues, greens, oranges, and reds, capturing her dance in a surreal juxtaposition of movement and stillness. Her skin tone, a light hue like the porcelain, adds an almost mystical quality to her form.*; **c**: *A high contrast portrait photo of a fluffy hamster wearing an orange beanie and sunglasses holding a sign that says "Let's PAINT!"*; **d**: *An extreme close-up of an gray-haired man with a beard in his 60s, he is deep in thought pondering the history of the universe as he sits at a cafe in Paris, his eyes focus on people offscreen as they walk as he sits mostly motionless, he is dressed in a wool coat suit coat with a button-down shirt , he wears a brown beret and glasses and has a very professorial appearance, and the end he offers a subtle closed-mouth smile as if he found the answer to the mystery of life, the lighting is very cinematic with the golden light and the Parisian streets and city in the background, depth of field, cinematic 35mm film.*; **e**: *poster of a mechanical cat, techical Schematics viewed from front and side view on light white blueprint paper, illustartion drafting style, illustation, typography, conceptual art, dark fantasy steampunk, cinematic, dark fantasy*; **f**: *I want to supplement vitamin c, please help me paint related food.*

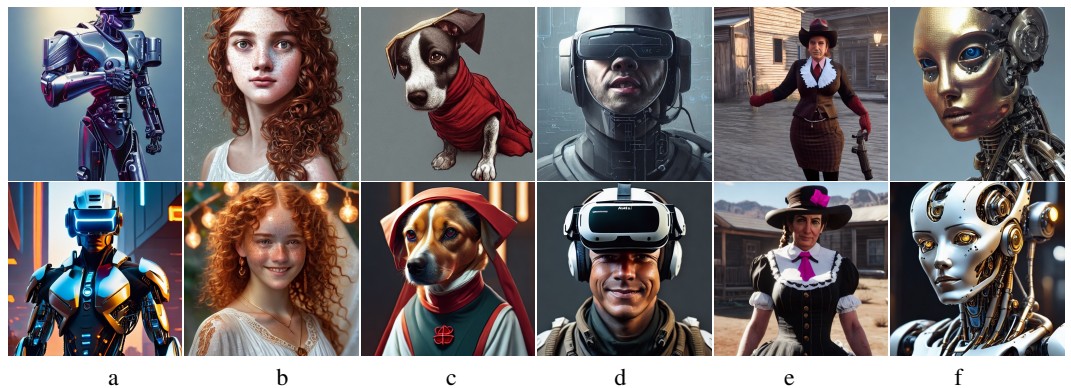

a          b          c          d          e          f

Figure 19: Top: Images generated from full-precision Stable Diffusion v1.5. Bottom: Images generated from BitsFusion. Prompts from left to right are: **a**: *new cyborg with cybertronic gadgets and vr helmet, hard surface, beautiful colours, sharp textures, shiny shapes, acid screen, biotechnology, tim hildebrandt, bruce pennington, donato giancola, larry elmore, masterpiece, trending on artstation, featured on pixiv, cinematic composition, dramatic pose, beautiful lighting, sharp, details, hyper - detailed, hd, hdr, 4 k, 8 k*; **b**: *portrait of teenage aphrodite, light freckles, curly copper colored hair, smiling kindly, wearing an embroidered white linen dress with lace neckline, intricate, elegant, mother of pearl jewelry, glowing lights, highly detailed, digital painting, artstation, concept art, smooth, sharp focus, illustration, art by wlop, mucha, artgerm, and greg Rutkowski*; **c**: *portrait of a dystopian cute dog wearing an outfit inspired by the handmaid ï¿½ s tale ( 2 0 1 7 ), intricate, headshot, highly detailed, digital painting, artstation, concept art, sharp focus, cinematic lighting, digital painting, art by artgerm and greg rutkowski, alphonse mucha, cgsociety*; **d**: *Portrait of a man by Greg Rutkowski, symmetrical face, a marine with a helmet, using a VR Headset, Kubric Stare, crooked smile, he's wearing a tacitcal gear, highly detailed portrait, scifi, digital painting, artstation, book cover, cyberpunk, concept art, smooth, sharp foccus ilustration, Artstation HQ*; **e**: *Film still of female Saul Goodman wearing a catmaid outfit, from Red Dead Redemption 2 (2018 video game), trending on artstation, artstationHD, artstationHQ*; **f**: *oil paining of robotic humanoid, intricate mechanisms, highly detailed, professional digital painting, Unreal Engine 5, Photorealism, HD quality, 8k resolution, cinema 4d, 3D, cinematic, professional photography, art by artgerm and greg rutkowski and alphonse mucha and loish and WLOP*

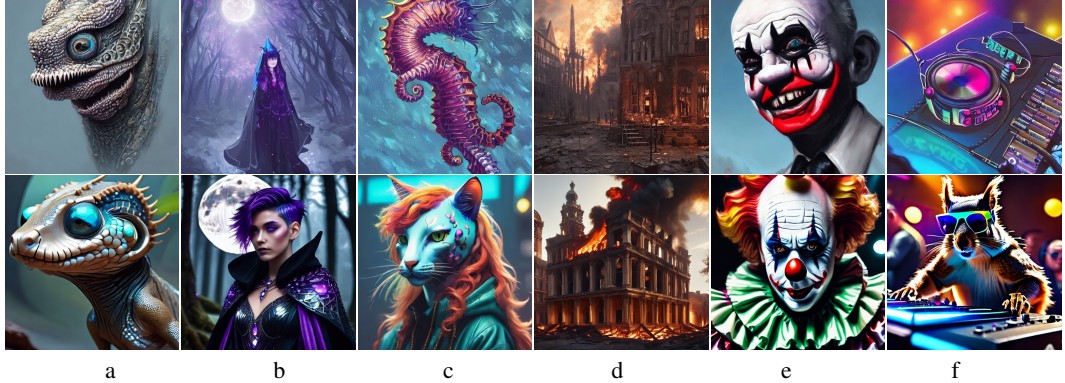

Figure 20: Top: Images generated from full-precision Stable Diffusion v1.5. Bottom: Images generated from BitsFusion. Prompts from left to right are: **a**: *anthropomorphic tetracontagon head in opal edgy darknimite mudskipper, intricate, elegant, highly detailed animal monster, digital painting, artstation, concept art, smooth, sharp focus, illustration, art by artgerm, bob eggleton, michael whelan, stephen hickman, richard corben, wayne barlowe, trending on artstation and greg rutkowski and alphonse mucha, 8 k*; **b**: *background shows moon, many light effects, particle, lights, gems, symmetrical!!! centered portrait dark witch, large cloak, fantasy forest landscape, dragon scales, fantasy magic, undercut hairstyle, short purple black fade hair, dark light night, intricate, elegant, sharp focus, digital painting, concept art, matte, art by wlop and artgerm and greg rutkowski and alphonse mucha, masterpiece*; **c**: *cat seahorse fursona, autistic bisexual graphic designer and musician, long haired attractive androgynous fluffy humanoid character design, sharp focus, weirdcore voidpunk digital art by artgerm, akihiko yoshida, louis wain, simon stalenhag, wlop, noah bradley, furaffinity, artstation hd, trending on deviantart*; **d**: *concept art of ruins of a victorian city burning down by j. c. leyendecker, wlop, ruins, dramatic, octane render, epic painting, extremely detailed, 8 k*; **e**: *hyperrealistic Gerald Gallego as a killer clown from outer space, trending on artstation, portrait, sharp focus, illustration, art by artgerm and greg rutkowski and magali Villeneuve*; **f**: *low angle photo of a squirrel dj wearing on - ear headphones and colored sunglasses, stadning at a dj table playing techno music at a dance club, hyperrealistic, highly detailed, intricate, smoke, colored lights, concept art, digital art, oil painting, character design by charlie bowater, ross tran, artgerm, makoto shinkai, wlop*

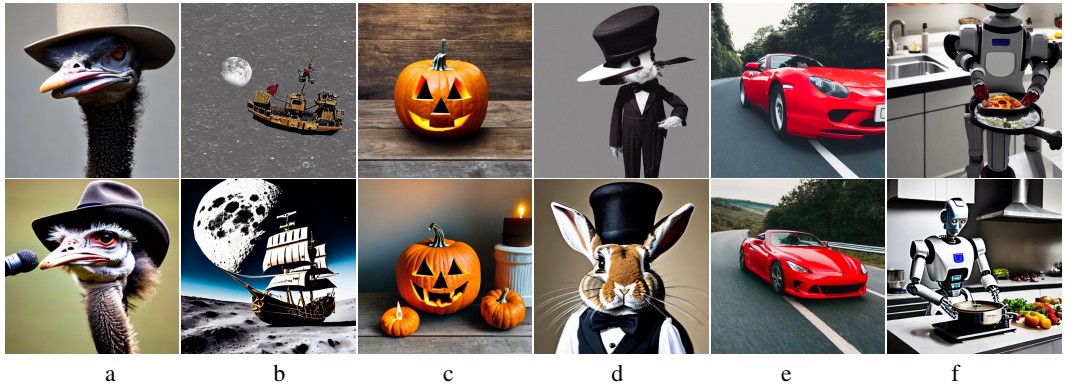

Figure 21: Top: Images generated from full-precision Stable Diffusion v1.5. Bottom: Images generated from BitsFusion. Prompts from left to right are: **a**: *a photograph of an ostrich wearing a fedora and singing soulfully into a microphone*; **b**: *a pirate ship landing on the moon*; **c**: *a pumpkin with a candle in it*; **d**: *a rabbit wearing a black tophat and monocle*; **e**: *a red sports car on the road*; **f**: *a robot cooking in the kitchen.*

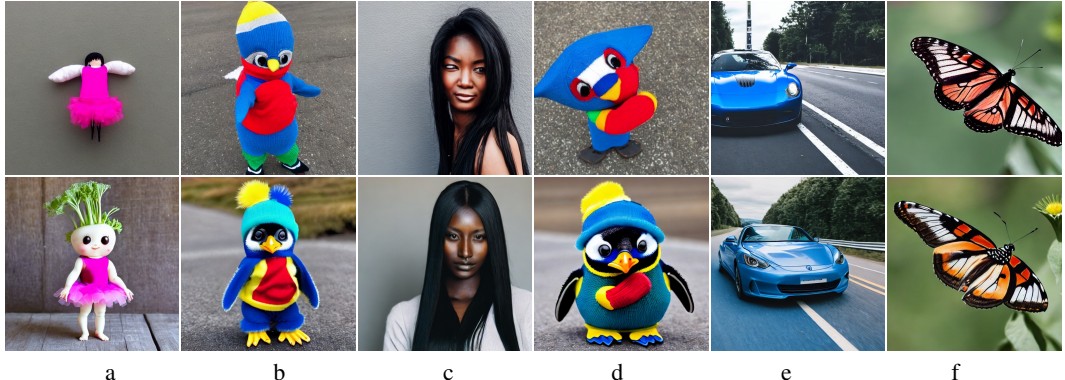

a  b  c  d  e  f

Figure 22: Top: Images generated from full-precision Stable Diffusion v1.5. Bottom: Images generated from BitsFusion. Prompts from left to right are: **a**: *a baby daikon radish in a tutu*; **b**: *a baby penguin wearing a blue hat, red gloves, green shirt, and yellow pants*; **c**: *a woman with long black hair and dark skin*; **d**: *an emoji of a baby penguin wearing a blue hat, red gloves, green shirt, and yellow pants*; **e**: *a blue sports car on the road*; **f**: *a butterfly.*

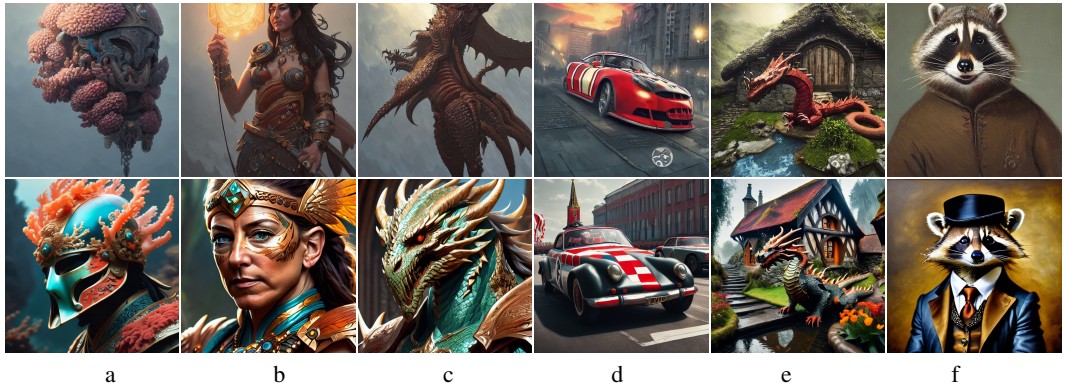

a  b  c  d  e  f

Figure 23: Top: Images generated from full-precision Stable Diffusion v1.5. Bottom: Images generated from BitsFusion. Prompts from left to right are: **a**: *Helmet of a forgotten Deity, clowing corals, extremly detailed digital painting, in the style of Fenghua Zhong and Ruan Jia and jeremy lipking and Peter Mohrbacher, mystical colors, rim light, beautiful lighting, 8k, stunning scene, raytracing, octane, trending on artstation*; **b**: *Jeff Bezos as a female amazon warrior, closeup, D&D, fantasy, intricate, elegant, highly detailed, digital painting, artstation, concept art, matte, sharp focus, illustration, hearthstone, art by Artgerm and Greg Rutkowski and Alphonse Mucha*; **c**: *Portrait of a draconic humanoid, HD, illustration, epic, D&D, fantasy, intricate, elegant, highly detailed, digital painting, artstation, concept art, smooth, sharp focus, illustration, art by artgerm and greg rutkowski and alphonse mucha, monster hunter illustrations art book*; **d**: *[St.Georges slaying a car adorned with checkered flag. Soviet Propaganda!!! poster!!!, elegant, highly detailed, digital painting, artstation, concept art, matte, sharp focus, illustration, octane render, unreal engine, photography]*; **e**: *a fire - breathing dragon at a medieval hobbit home, ornate, beautiful, atmosphere, vibe, mist, smoke, chimney, rain, wet, pristine, puddles, waterfall, clear stream, bridge, forest, flowers, concept art illustration, color page, 4 k, tone mapping, doll, akihiko yoshida, james jean, andrei riabovitchev, marc simonetti, yoshitaka amano, digital illustration, greg rutowski, volumetric lighting, sunbeams, particles*; **f**: *portrait of a well-dressed raccoon, oil painting in the style of Rembrandt*

