# OpenReview forum: "BitsFusion: 1.99 bits Weight Quantization of Diffusion Model"
_NeurIPS.cc/2024/Conference — NeurIPS 2024 poster_

### Official Review · Reviewer_Fq1C · 2024-06-29

**Soundness:** 3
**Presentation:** 3
**Contribution:** 2
**Rating:** 6
**Confidence:** 4

**Summary:**

This paper proposes a weight quantization method to quantize the UNet of SDv1.5 to 1.99 bits while maintaining model performance comparable to the floating-point model. The approach includes a series of techniques such as bit-width allocation for mixed-precision quantization, low-bit diffusion model initialization, and a two-stage training pipeline. Extensive experiments demonstrate the effectiveness of the proposed method. Results show that the quantized diffusion model can even outperform the floating-point SDv1.5.

**Strengths:**

1. The paper is well-written and easy to follow.
2. The per-layer sensitivity analysis is comprehensive, considering both quality and contextual information, which is helpful for future research.
3. The effectiveness of the proposed approach is validated across multiple datasets and evaluation metrics.
4. The generated results look impressive.

**Weaknesses:**

1. It is well-known that QAT-based approaches can achieve superior performance at the expense of additional training effort. While this work involves training the weights via QAT, it does not include comparisons with other QAT-based methods such as Q-DM and TDQ.
2. It is unclear how the sensitivity threshold in Section 3 is determined and whether this hyperparameter generalizes to other models.

**Questions:**

1. The setting of activation is not mentioned in the paper. Are the activations kept in floating point?
2. What is the dataset used for stage-II training? Suppose the training set of the corresponding dataset is used. In that case, claiming the model outperformed the floating-point SDv1.5 model may be inappropriate since the SDv1.5 model is evaluated under a zero-shot setting.
3. The average bits uses $\log(2^n + 1)$ bits for calculation. Is it necessary to incorporate an optimal encoding method such as Huffman encoding to reach this average bit-width?

**Limitations:**

See weaknesses and questions.

---

> ### Author Rebuttal · Authors · 2024-08-06
>
> **Q1. Comparison with QAT-based approaches such as Q-DM and TDQ.**
>
> A1. Thanks for the suggestion. We have compared the QAT-based approach LSQ and EfficientDM in Table 3 of the main paper. Here, we provide more results for Q-DM and TDQ on PartiPrompts with a CFG scale of 7.5. Compared to Q-DM with 2-bit weight quantization, our BitsFusion achieves a CLIP score of 0.3212, which is 0.228 higher than Q-DM. For a fair comparison with TDQ, which optimizes activation quantization through dynamic quantization, we also apply 8-bit activation quantization to our 1.99-bit BitsFusion. As shown in the table below, our BitsFusion achieves a CLIP score of 0.3193, surpassing TDQ with a score of 0.2907.
>
> >| Methods | Weight Bit-width | Activation Bit-width | CLIP score |
> | :---- | :---- | :---- | :---- |
> | Stable Diffusion v1.5 | 32 | 32 | 0.3175 |
> | Q-DM | 2 | 32 | 0.2984 |
> | BitsFusion | 1.99 | 32 | 0.3212 |
> | TDQ | 2 | 8 | 0.2907 |
> | BitsFusion | 1.99 | 8 | 0.3193 |
>
> ---
>
> **Q2. How to determine the sensitivity threshold in Section 3.**
>
> A2. Thanks for the comment. We empirically set the sensitivity threshold to achieve an average of 1.99 bits. The sensitivity threshold only affects the average bits. It can be adjusted as needed for other models with different bit requirements, thus it is generalized to other models.
>
> We would like to kindly note here the sensitive threshold is not a hyperparameter that influences the model training. Thanks for mentioning this. We will modify our manuscript accordingly to emphasize this distinction.
>
> ---
> **Q3. Activation quantization.**
>
> A3. Thanks for the valuable question. Since this paper is focusing on reducing the model size, we explore the weight quantization on the diffusion model and keep activation as floating point value. However, 8-bit activation quantization can be directly applied to our approach. Based on 1.99 bits weight quantization, we further apply 8-bit activation quantization to all layers except for the first and last convolutional layer and train the corresponding scaling factors. The experimental settings are the same as those described in Section 5 of the main paper. With a CFG scale of 7.5, our stage-I model and stage-II model can achieve CLIP scores of 0.3172 and 0.3193 on the PartiPrompts, respectively. For the TIFA score, our stage-I model and stage-II model can achieve 0.786 and 0.805 with a CFG scale of 7.5. With two-stage training, our BitsFusion model with a 1.99-bit weight and 8-bit activation quantization can still outperform the full-precision SDv1.5.
>
> >| Methods | Weight Bit-width | Activation Bit-width | CLIP score | TIFA |
> | :---- | :---- | :---- | :---- | :---- |
> | Stable Diffusion v1.5 | 32 | 32 | 0.3175 | 0.788 |
> | Stage-I | 1.99 | 8 | 0.3172 | 0.786 |
> | Stage-II | 1.99 | 8 | 0.3193 | 0.805 |
>
> ---
> **Q4. Datasets used in the stage-II training.**
>
> A4. Thanks for the question. Our training data consists of synthetic data and real images, and there is no overlap between the training data and evaluation data. To clarify, our model performs *zero-shot evaluation*, which is the same as SDv1.5. Specifically, we perform different evaluations to verify the effectiveness of BitsFusion, such as CLIP scores on the MS-COCO dataset and PartiPrompts, GenEval score, and TIFA score. PartiPrompts provides prompts from different categories. GenEval provides prompts to evaluate compositional image properties such as object co-occurrence, position, count, and color. TIFA provides various prompts and uses the VQA model to measure the text-to-image model if it can accurately answer the questions. Evaluating CLIP scores, TIFA score, and GenEval score do not include any reference images, and we do not have the same prompts from these datasets in the training.
>
> ---
> **Q5. Is it necessary to incorporate an optimal encoding method such as Huffman encoding to reach the averaged bit-width?**
>
> A5. Thanks for the comment. We calculate the average bits according to Section I in the Appendix with $\frac{\sum_{i} log \(2^{b_{i}^*}+1\)* N_i + 16*N_{tf}}{N_w}$, where $b_{i}^{\ast}$ is the calculated bit-width in the $i_{th}$ layer, $N_i$ is the number of weights of the $i_{th}$ layer, $N_{tf}$ is the number of parameters for pre-cached time features, and $N_w$ is the total number of weights in linear and convolutional layers.  We do not need the Huffman encoding to reach this average bit-width. However, as suggested by the reviewer, Huffman encoding is a valuable tool to further reduce our storage size.

---

> > ### Author Response · Authors · 2024-08-10
> >
> > Dear Reviewer Fq1C,
> >
> > Thank you very much for your valuable feedback and the positive evaluation of our work.  We have included detailed explanations in response to your questions. As the deadline for the discussion period approaches, we would appreciate your review of these explanations to confirm that they fully meet your expectations and resolve any remaining concerns. Thank you once again for your insightful contributions.
> >
> > Best regards,
> >
> > The Authors

---

> > > ### Comment · Reviewer_Fq1C · 2024-08-11
> > >
> > > I thank the authors for their response, which addresses most of my concerns, and I would like to maintain my positive rating. However, I still have one question regarding the calculation of average bits. The balanced integer should require an additional bit to avoid overflow, which would mean the average bit calculation should be $log(2^n)+1$ instead of $log(2^n+1)$. The term $log(2^n+1)$ implicitly suggests the minimum bit required for representing the amount of information, which would typically require an optimal encoding approach to achieve this average bit. I am still unclear about how this can be achieved without additional encoding.

---

> > > > ### Author Response · Authors · 2024-08-12
> > > >
> > > > Dear Reviewer Fq1C,
> > > >
> > > > Thank you for your comments. The “averaged 1.99 bits” refers to the theoretical average for linear and convolutional layers according to previous quantization works [A, B]. In practical deployment, we use group encoding to encode multiple values simultaneously, closely achieving the theoretical bits.
> > > >
> > > > Here we provide more details with a simple example of three symbols {-1, 0, 1}, where the theoretical bit-width is log(3) = 1.58 bits. To approach this in practice, we encode blocks of 8 values—each set of three ternary values representing $3^8 = 6561$ unique combinations. We encode these 6561 combinations using $\lceil log_2(6561) \rceil = 13$ bits, resulting in an average of 13/8 = 1.625 bits per value, closely aligning with the theoretical 1.58 bits.
> > > >
> > > > This group encoding technique is used throughout our quantized text-to-image diffusion model, leading to a total model size of 219 MB as indicated in line 39 of our main paper, which is 7.9$\times$ reduction compared with the fp16 model, aligning with the theoretical size saving.
> > > >
> > > > We believe Huffman encoding is another valuable tool, which is orthogonal to our approach, for closely achieving the theoretical bits and reducing storage size. Thank you for suggesting this useful approach and for your positive response!
> > > >
> > > > Best,
> > > >
> > > > Authors
> > > >
> > > > ---
> > > >
> > > > References:
> > > >
> > > > [A] Ma, Shuming, et al. “The era of 1-bit llms: All large language models are in 1.58 bits.” arXiv preprint arXiv:2402.17764 (2024).
> > > >
> > > > [B] Li, Xiuyu, et al. “Q-diffusion: Quantizing diffusion models.” CVPR, 2023.

---

> > > > > ### Comment · Reviewer_Fq1C · 2024-08-13
> > > > >
> > > > > Thank you for providing more details regarding the calculation of average bits, which addresses my question well. I suggest that the authors incorporate this discussion into the paper.

---

> > > > > > ### Author Response · Authors · 2024-08-14
> > > > > >
> > > > > > Thank you for your response and positive rating! We're glad to hear that your concerns have been addressed. We will incorporate the related discussion into the paper.

---

### Official Review · Reviewer_W2xs · 2024-07-06

**Soundness:** 3
**Presentation:** 3
**Contribution:** 3
**Rating:** 6
**Confidence:** 5

**Summary:**

The paper presents BitsFusion, a novel method for weight quantization of diffusion models, specifically applied to the UNet architecture in Stable Diffusion v1.5. The approach quantizes weights to 1.99 bits, achieving a model size reduction of 7.9 times while enhancing or maintaining image generation quality. The experiment evaluates the quantized model across various benchmarks, including MS-COCO, TIFA, GenEval, and human evaluations, and demonstrates superior performance of the quantized model compared to full-precision Stable Diffusion v1.5.

**Strengths:**

1.The paper is well-organized and presents its content in an accessible manner. The organization of the paper is clear.

2.This paper mainly focuses on quantizing large-scale models like Stable Diffusion v1.5 with fewer bits than 4, which has significant implications for the industry and has achieved satisfactory results.

3.The paper methodologically addresses the challenges associated with low-bits quantization in SD and provides a robust solution through the BitsFusion framework. The experiments demonstrate the effectiveness of BitsFusion in achieving superior performance compared to existing methods on various large-scale datasets.

**Weaknesses:**

1.The motivation of the paper seems to be heuristic and lacks necessary theoretical analysis, but the impact of this method in the industry is worth looking forward to.

**Questions:**

What's the total resource consumption for this entire process? For instance, how many GPUs with what memory capacity were utilized, and for how many hours was the training conducted? How does this compare to the resources required for a baseline that trains the original SD v1.5?

**Limitations:**

See weakness.

---

> ### Author Rebuttal · Authors · 2024-08-06
>
> **Q1. About the motivation of this paper.**
>
> A1. The overall motivation for performing quantization on the diffusion model is to reduce significant burdens for transferring and storing the model due to its large size. To this end, we propose several methods with corresponding motivations for model quantization:
>
> Mixed-precision Strategy:
> - Mixed-precision Strategy: Various layers exhibit different sensitivities during quantization. To achieve a lower quantization error, it is crucial to employ a mixed-precision approach where more sensitive layers are quantized at higher bit-widths to retain essential information, while less sensitive layers can be quantized at lower bit-widths. In Section 3.2, we conduct the quantization error of layers for different bit-widths via appropriate metrics and obtain valuable observations to guide our mixed-precision approach.
>
> Initialization Schemes:
> - Time Embedding Pre-computing and Caching: During the inference, the embedding for a fixed time step is always the same (Line 177\~179). Therefore, we can pre-compute the embedding offline and load cached values during inference, instead of computing the embedding every time.
> - Adding Balance Integer: We observe that weight distributions in the diffusion model are symmetric around zero. However, the existing quantization works disrupt this symmetric property (Line 182\~189). Therefore, we introduce an additional value to balance the original values to maintain this property.
> - Scaling Factor Initialization via Alternating Optimization: Min-Max initialization leads to a large quantization error and the increased difficulty to converge in extremely low-bit quantization settings like 1-bit (Line 198\~201). Therefore, we propose using an alternating optimization method to improve scaling factor initialization.
>
> Training Schemes:
> - CFG-aware Quantization Distillation: Distillation is an effective approach to recover the performance degradation caused by quantization. Additionally, considering that CFG enhances the performance of the diffusion model, we leverage CFG-aware distillation to effectively recover the quantization error (Line 217\~218).
> - Feature Distillation: To further improve the generation quality of the quantized model, we distill the full-precision model at a more fine-grained level (Line 220\~221). Therefore, we adopt the feature distillation.
> - Quantization Error-aware Time Step Sampling: We analyze the quantization error during various time steps during training. As shown in Fig. 4 in the main paper, the quantization error does not distribute equally across all time steps in the quantized model (Line 230\~240). Therefore, we propose a quantization error-aware time step sampling based on Beta distribution to reduce the overall quantization error.
>
> We believe that our comprehensive analysis and the proposed methods will contribute significantly to the field of quantization on diffusion models.
>
> ---
>
> **Q2. About training computation.**
>
> A2. Thank you for the suggestions. Here we provide more details for our two-stage training pipeline. For the stage-I training, we use 8 NVIDIA A100 GPUs with 40GB memory, and a total batch size of 256 to train the quantized model for 20K iterations. The total training time is within 40 hours. For the stage-II training, we use 32 NVIDIA A100 GPUs with 40GB memory, and a total batch size of 1024 to train the quantized model for 50K iterations. The total training time is within 100 hours.
>
> Note that the stage-II training aims to further improve the performance of the quantized model such that it is better than the full-precision Stable Diffusion v1.5. If reducing the training cost would be the goal, we can remove stage-II training and only use stage-I training. Stage-I training is able to achieve similar performance as the full-precision Stable Diffusion v1.5, as shown in Fig. 5 of the main paper.
>
> To better understand the training cost of our model, we show the training cost comparison between our model and Stable Diffusion v1.5 (the training cost is obtained from the public resource https://huggingface.co/runwayml/stable-diffusion-v1-5). We use the total trained samples to compare the training costs as the training hours depending on the actual hardware used. As shown in the following table, our stage-I training only requires 0.16% training cost compared with training the Stable Diffusion v1.5 from scratch, and stage-II training requires 1.6% training cost compared with training the Stable Diffusion v1.5 from scratch. For the training time, based on the public information (Huggingface), SD-v1.5 is trained for 30-60 days on 256 GPUs, while our model takes 5~6 days on at most 32 GPUs.
>
> >| Stage | A100 | Batch size | Gradient accumulation | Total batch size | 256x256 iters (K) | 512x512 iterations (K) | Total training samples (M) | Percentage of total training samples compared to SD-v1.5 |
> | :---- | :---- | :---- | :---- | :---- | :---- | :---- | :---- | :---- |
> | Stable Diffusion v1.5 | 256 | 4 | 2 | 2048 | 237 | 1304 | 3155.97 | 100% |
> | Stage-I | 8 | 8 | 4 | 256 | \- | 20 | 5.12 | 0.16% |
> | Stage-II | 32 | 8 | 4 | 1024 | \- | 50 | 51.2 | 1.6% |

---

> > ### Comment · Reviewer_W2xs · 2024-08-08
> >
> > Thank you for addressing my concerns regarding the training computation. I will keep my scores as wa.

---

> > > ### Author Response · Authors · 2024-08-08
> > > **Thank you for the response**
> > >
> > > Thank you for your quick response and positive rating! We're pleased to hear that your concerns have been addressed. We will incorporate the relevant discussion in the paper.

---

### Official Review · Reviewer_tjXw · 2024-07-13

**Soundness:** 2
**Presentation:** 3
**Contribution:** 2
**Rating:** 5
**Confidence:** 5

**Summary:**

This paper demonstrates a quantized diffusion model called BitFusion, which successfully quantize the Stable Diffusion (SD) v1.5 to 1.99 bits with 7.9x smaller size.  They first analyse the SD model in a layer perspective and assign the optimal bit based on the analysis. Then they propose a training pipeline to perform quantization-aware training (QAT) based on the analysis result. The experiment result shows that the quantized diffusion model BitFusion outperforms the full-precision SD v1.5 model.

**Strengths:**

1. The result is significant. BitFusion successfully compressed the SD model while maintaining its performance.

2. The authors successfully combine novel quantization techniques and existing quantization techniques, showing an effective way to quantize SD model to extreme low bit.

3. The paper provides valuable observations about different layers' behaviour under quantization.

**Weaknesses:**

1. The analysis and experiment according to scaling factor initialization is not comprehensive. The authors propose alternating optimization for scaling factor initialization which is based on the MSE between quantized weight and full-precision weight. However, quantized weight initialization is well-studied and there are existing weight initialization technique that optimize the same objective. Nevertheless, there are some advanced initialization technique like adaround [1]. Those techniques are adapted by other Diffusion Model quantization research [2]. An experiment to compare the alternating Opt. and those initialization techniques is necessary.

2. Stable Diffusion 1.5 is no longer the SOTA in Diffusion text-to-image model. Diffusion transformer (DiT) is the new trend in diffusion model research. The conclusion on UNet-based diffusion model like SD 1.5 might not be precise for the DiT diffusion model.

3. FID is widely adapted by most of the diffusion model quantization research. Although the authors provide human evaluation and justify that FID is not accurate compared with human evaluation, the FID comparison between BitFusion and other baseline models should be included in the main body of the paper.

4. In this paper [3], the authors propose a 1.58-bit LLM quantization approach comparable to Bit Fusion at model size reduction. However, they can perform activation quantization to 8-bit while Bit Fusion doesn't.


[1] Nagel, Markus, et al. "Up or down? adaptive rounding for post-training quantization." International Conference on Machine Learning. PMLR, 2020.

[2] Li, Xiuyu, et al. "Q-diffusion: Quantizing diffusion models." Proceedings of the IEEE/CVF International Conference on Computer Vision. 2023.

[3] Ma, Shuming, et al. "The era of 1-bit llms: All large language models are in 1.58 bits." arXiv preprint arXiv:2402.17764 (2024).

**Questions:**

NA

**Limitations:**

The authors honestly point out their limitation: not quantizing VAE and text encoder and not quantizing activation. For the first limitation, it is quite common that diffusion model quantization research doesn't quantize VAE and text encoder according to the literature. But for the second limitation, the authors leave it for future work and doesn't address it. Activation quantization is recommended, as the fact that activation usually has higher quantization difficulty.

---

> ### Author Rebuttal · Authors · 2024-08-06
>
> **Q1. Comparison with adaround \[1\], which is used in Diffusion Model quantization research \[2\]**.
>
> A1. Thank the reviewer for suggesting the adaround, which is a relevant method \[1\]. However, we would like to kindly emphasize that adaround \[1\] focuses on *post-training quantization* by proposing a better weight-rounding operation, while our scaling factor initialization is used for *training-aware quantization*. These two technologies are applied during different quantization pipelines. As highlighted by Reviewer WzcG, our scaling factor initialization is novel.
>
> Indeed, as mentioned by the reviewer, the Q-Diffusion \[2\] uses the adaround, implemented by MinMax and Brute-force MSE searching. As analyzed in the paper (Line 198\~201), MinMax is able to maintain the outlier weights but overlook the overall quantization error in the extreme low-bit setting, thus sacrificing the performance of the quantized model. To further prove this, we have conducted an ablation analysis in Table 4 of the main paper. With the alternating optimization (indicated by the row labeled “+Alternating Opt.”), the CLIP score is improved from 0.3054 (indicated by the row labeled “+Balance”, which uses MinMax as the default initialization method) to 0.3100.
>
> As for the brute-force MSE, it takes lots of time to find the ideal scaling factor than our initialization approach. Following the same setting, we measure the CLIP score where the brute-force MSE in Q-Diffusion \[2\] can achieve a score of 0.3098, while our alternating optimization achieves 0.3100. However, the brute-force MSE requires 13 minutes to initialize the scaling factor by traversing 80 different scaling factors in each layer. In contrast, our alternating optimization significantly reduces the initialization time, which only needs 5 minutes with 10 optimization iterations.
>
> In fact, we have compared our approach with Q-Diffusion \[2\]. As shown in Table 3 of the main paper, our 1.99 bits model can achieve a 0.3175 CLIP score, even outperforming the 4 bits Q-Diffusion model which has a CLIP score of 0.3137 evaluated on the PartiPrompts.
>
> Thanks again for recommending the comparison with adaround. We will revise the manuscript to include the above discussion.
>
> ---
>
> **Q2. Quantization of the diffusion transformer (DiT).**
>
> A2. We agree with the reviewer that besides UNet, DiT is another promising architecture used as the backbone for the diffusion model. This work targets the quantization of UNet for several reasons.
>
> - First, UNet is a widely adopted architecture for diffusion models. Besides the text-to-image generation, there are numerous applications like ControlNet and IP-Adapter that are built upon the UNet-based diffusion models. However, there is no previous effort to study the extremely low-bit quantization for such an architecture. Therefore, it is important to understand how to quantize the UNet to low-bits.
> - Second, UNet is more efficient than DiT during inference, especially on resource-constrained devices. Using quantization to compress the efficient architecture has important practical values, as agreed by Reviewer WzcG and W2xs. Even if we apply the quantization to DiT models, how to deploy such models on resource-constrained devices is unclear, *e.g.*,  we are not able to run DiT-based text-to-image models on the iPhone.
> - Third, although there is quite some work studying the improvement of DiT-based diffusion models, the UNet-based model still shows competitive performance for text-to-image generation. For instance, the open-sourced Kolors ([https://github.com/Kwai-Kolors/Kolors/tree/master?tab=readme-ov-file\#-evaluation](https://github.com/Kwai-Kolors/Kolors/tree/master?tab=readme-ov-file\#-evaluation)), which is built upon SDXL, achieves the SOTA results for text-to-image generation.
>
> With all that being said, we believe that our approach can be applied to quantize DiT models, especially considering the quantization error analysis and training pipelines proposed in this paper are generic methods.
>
> Thanks again for suggesting the quantization of DiT, which is an interesting topic that we would like to explore by using our proposed methods.
>
> ---
> **Q3. Adding the FID comparison to the main paper.**
>
> A3. Thanks for the suggestion. We will move the FID comparison from the Appendix to the main paper.
>
> ---
>
> **Q4. About the activation quantization for the 1.58-bit LLM quantization approach.**
>
> A4. This work mainly targets the size reduction of the diffusion model. Therefore, we did not show the activation quantization. Nevertheless, 8-bit activation quantization can be directly applied to our approach. Based on 1.99 bits weight quantization, we further apply 8-bit activation quantization to all layers except for the first and last convolutional layer and train the corresponding scaling factors. The experimental settings are the same as those described in Section 5 of the main paper. With a CFG scale of 7.5, our stage-I model and stage-II model can achieve CLIP scores of 0.3172 and 0.3193 on the PartiPrompts, respectively. For the TIFA score, our stage-I model and stage-II model can achieve 0.786 and 0.805 with a CFG scale of 7.5. With two-stage training, our BitsFusion model with a 1.99-bit weight and 8-bit activation quantization can still outperform the full-precision SD-v1.5.
>
> >| Methods | Weight Bit-width | Activation Bit-width | CLIP score | TIFA score |
> | :---- | :---- | :---- | :---- | :---- |
> | Stable Diffusion v1.5 | 32 | 32 | 0.3175 | 0.788 |
> | Stage-I | 1.99 | 8 | 0.3172 | 0.786 |
> | Stage-II | 1.99 | 8 | 0.3193 | 0.805 |
>
> ---
>
> \[1\] Nagel, Markus, et al. "Up or down? adaptive rounding for post-training quantization." International Conference on Machine Learning. PMLR, 2020\.
> \[2\] Li, Xiuyu, et al. "Q-diffusion: Quantizing diffusion models." Proceedings of the IEEE/CVF International Conference on Computer Vision. 2023\.

---

> > ### Author Response · Authors · 2024-08-10
> >
> > Dear Reviewer tjXw,
> >
> > Thank you very much for your insightful comments. In response to your comments, we have provided detailed explanations to clarify the points discussed and address the concerns you highlighted. As the deadline for the discussion period is quickly approaching, we are keen to know if our responses meet your expectations and have addressed all your concerns effectively. Thank you again for your valuable time and expertise.
> >
> > Best regards,
> >
> > The Authors

---

> > > ### Comment · Reviewer_tjXw · 2024-08-11
> > > **Thank you for your response**
> > >
> > > Dear authors,
> > >
> > > Thanks for your thorough classification and explanation. Although some concerns like Quantization for DiT models require further exploration, I will raise my score.

---

> > > > ### Author Response · Authors · 2024-08-12
> > > >
> > > > Dear Reviewer tjXw,
> > > >
> > > > Thank you for raising your score. We are glad to know that our response addresses most of your concerns.
> > > >
> > > > We would like to use this opportunity to kindly clarify that the further exploration of DiT quantization using our approach is not a limitation or weakness of this paper. Exploring the DiT quantization is beyond the scope of this work.
> > > >
> > > > As we mentioned in the answer of Q2, UNet is more suitable for the resource-constrained hardware, which is the hardware that quantization is deployed the most. Additionally, with all due respect, we would like to kindly argue that, DiT is not necessarily better than UNet. For example, in the recent work [A], people show that under the same training dataset, schedule, hardware, and training time, the UNet still achieves better performance than DiT (as can be seen in Fig. 6). Therefore, we would like to kindly mention that the value of this work should not be underplayed due to the reason of missing DiT quantization.
> > > >
> > > > Best,
> > > >
> > > > Authors
> > > >
> > > > ---
> > > >
> > > > References:
> > > >
> > > > [A] Li, Hao, et al. “On the Scalability of Diffusion-based Text-to-Image Generation”. CVPR, 2024.

---

> ### Comment · Reviewer_tjXw · 2024-08-13
> **Additional Comment ny authors**
>
> Dear authors,
>
> Thanks for your valuable information. It is worth noting that the findings in prior diffusion model quantization research are mostly based on the U-Net structure, like shortcut splitting quantization in Q-Diffusion [1] and temporal information aware reconstruction in TFMQ-DM [2]. However, the findings in BitFusion, like how to decide the optimal precision, are not limited to the U-Net diffusion model. So I still suggest BitFusion's effectiveness be further verified with DiT as most of the SOTA diffusion models are DiT-based.
>
> Best,
> Reviewer tjXw
>
> [1] Li, Xiuyu, et al. "Q-diffusion: Quantizing diffusion models." Proceedings of the IEEE/CVF International Conference on Computer Vision. 2023.
> [2] Huang, Yushi, et al. "Tfmq-dm: Temporal feature maintenance quantization for diffusion models." Proceedings of the IEEE/CVF Conference on Computer Vision and Pattern Recognition. 2024.

---

> > ### Author Response · Authors · 2024-08-14
> > **Response to additional comments**
> >
> > Dear Reviewer tjXw,
> >
> > Thank you for your additional response. We appreciate the time and effort you have invested in this paper. Please allow us to summarize our agreements and disagreements regarding the DiT quantization.
> >
> > **Agreements**:
> >
> > - The proposed quantization methods in BitsFusion are very generic and can be applied to other architectures, like DiT.
> >
> > **Disagreements**:
> >
> > - The reviewer believes DiT is the SOTA diffusion model.
> > - The reviewer mentioned that, since DiT is the SOTA diffusion model, lacking experiments of DiT is a limitation or concern for this work.
> >
> > Here, we would like to kindly present some **facts** that support us to humbly disagree with the reviewer.
> >
> > **First**, as we mentioned in the previous response, **we did not see any paper clearly showing that DiT is better than UNet**. On the contrary, the recent work [A] shows that under the same training dataset, schedule, hardware, and training time, the **UNet is better than DiT** (as seen in Fig. 6 in [A]).
> >
> > **Second**, we understand some recent DiT-based works like SD3 [B] show promising text-to-image generation. However, the architecture in SD3 is more complicated than the original DiT. Additionally, the training recipe and noise scheduling are different from those used to train the UNet-based models. Even if we take the training dataset, recipe, and noise scheduling aside and only focus on the comparison of the final generated images, the open-sourced Kolors (as mentioned in the previous response), which is built upon SDXL, achieves the SOTA results for text-to-image generation. The Kolors demonstrates that the **UNet is better than SD3 (DiT-based model)**.
> >
> > **Third**, taking a step back, we are still interested in applying our approach to DiT. However, it is non-trivial to achieve comparable results since DiT has a different architecture than the UNet-based model. Moreover, there might be some other new observations and problems in DiT-based models that we need to solve when doing quantization for them. Therefore, exploring the DiT-based model belongs to another research direction and we believe such experiments are beyond the scope of this paper.
> >
> > We humbly hope the above arguments can make sense to the reviewer.
> >
> > Best,
> >
> > Authors
> >
> > ---
> >
> > References:
> >
> > [A] Li, Hao, et al. “On the Scalability of Diffusion-based Text-to-Image Generation”. CVPR, 2024.
> >
> > [B] Esser, Patrick, et al. “Scaling Rectified Flow Transformers for High-Resolution Image Synthesis”. ICML, 2024.

---

### Official Review · Reviewer_WzcG · 2024-07-13

**Soundness:** 3
**Presentation:** 4
**Contribution:** 3
**Rating:** 6
**Confidence:** 4

**Summary:**

This paper proposes a novel weight quantization method called "BitsFusion" for compressing the Stable Diffusion v1.5 model. The primary goal is to address the issue of large model sizes, which hinder the deployment of diffusion models on resource-constrained devices. The BitsFusion framework quantizes the UNet component of Stable Diffusion v1.5 to 1.99 bits, achieving a 7.9× reduction in model size while improving image generation quality. The approach involves mixed-precision quantization, novel initialization techniques, and an improved two-stage training pipeline. Extensive experiments on various benchmark datasets demonstrate that the quantized model outperforms the original full-precision model in terms of generation quality and text-image alignment. My detailed comments are as follows.

**Strengths:**

1. The BitsFusion framework is an innovative approach to compressing the UNet component of Stable Diffusion v1.5 to 1.99 bits, achieving a significant reduction in model size while enhancing image quality by using mixed-precision quantization, novel initialization techniques, and an improved two-stage training pipeline.
2. The paper provides a thorough analysis of quantization errors and develops a mixed-precision strategy based on this analysis, which contributes to the theoretical understanding of low-bit quantization for large-scale diffusion models.
3. The results demonstrate the effectiveness of the proposed framework. The quantized model consistently outperforms the original full-precision model across various metrics, including CLIP score, TIFA score, and GenEval score. The model's ability to achieve a 7.9× reduction in size while maintaining or improving performance highlights its practical potential for deployment on resource-constrained devices.
4. The manuscript is well-written and easy to understand, with clear explanations of the methodology and experimental procedures. Sufficient experimental details are provided, ensuring that the results can be reproduced by other researchers, which enhances the paper's credibility and utility.

**Weaknesses:**

1. The two-stage training pipeline, while effective, may introduce additional computational complexity and training time, which could be a drawback for some applications. However, the paper does not explicitly discuss the additional computational complexity and training time it introduces. To provide a more comprehensive evaluation of this method, it is recommended that the authors discuss the potential additional computational overhead introduced by the two-stage training pipeline and its impact on training time.
2. The study demonstrates that there is a high correlation between certain metrics, such as MSE and PSNR, and uses these correlations to validate the effectiveness of their quantization methods. However, correlation does not imply causation. The study does not delve into why these metrics are correlated or explain the intrinsic mechanisms that cause certain layers to be more sensitive to quantization (e.g., why quantization affects some layers more than others, and what intrinsic properties of these layers make them more prone to quantization-induced errors). Understanding these mechanisms is crucial for improving the quantization process, as knowing why certain layers are sensitive can lead to more targeted and effective quantization strategies.
3. The claim that the 1.99-bit quantized model outperforms the full-precision model in all evaluated metrics is a significant overgeneralization. This assertion, based on specific experimental settings and datasets, may not fully represent the diversity encountered in real-world applications. Primarily, the evaluation on the MS-COCO 2014 validation set, though extensive, does not cover a wide variety of prompts, limiting the assessment of the model's ability to handle different types of inputs (e.g., complex scenes, abstract concepts, specific artistic styles). Additionally, the tests do not include various real-world conditions such as different lighting, backgrounds, or object complexities. Therefore, the claim that the 1.99-bit quantized model outperforms the full-precision model in all evaluated metrics is premature. A more robust evaluation including varied datasets, prompts, and real-world conditions, such as [A-E],  is necessary to substantiate such a broad claim.
4. The discussion regarding model compression techniques is insufficient. It would be better for the authors to present more model quantization methods [F-J].

[A] Visual Genome: Connecting Language and Vision Using Crowdsourced Dense Image Annotations. International Journal of Computer Vision. IJCV. 2017.
[B]  Flickr30k Entities: Collecting Region-to-Phrase Correspondences for Richer Image-to-Sentence Models. ICCV. 2015.
[C]  The Open Images Dataset V4: Unified Image Classification, Object Detection, and Visual Relationship Detection at Scale. IJCV. 2020.
[D]  Conceptual 12M: Pushing Web-Scale Image-Text Pre-Training To Recognize Long-Tail Visual Concepts. CVPR. 2021.
[E]  Scene Parsing through ADE20K Dataset. CVPR. 2017.
[F] Binary quantized network training with sharpness-aware minimization. Scientific Computing 2023.
[G] Network Quantization with Element-Wise Gradient Scaling. CVPR 2021.
[H] Single-path bit sharing for automatic loss-aware model compression. TPAMI 2023.
[I] Generative Data Free Model Quantization with Knowledge Matching for Classification. TCSVT 2023.

**Questions:**

Please check the weakness section.

---

> ### Author Rebuttal · Authors · 2024-08-06
>
> **Q1. About training computation.**
>
> A1. Thanks for the suggestions. For the stage-I training, we use 8 NVIDIA A100 GPUs with a total batch size of 256 to train the quantized model for 20K iterations. The training time is within 40 hours.
> For the stage-II training, we use 32 NVIDIA A100 GPUs with a total batch size of 1024 to train the quantized model for 50K iterations. The total time is within 100 hours.
>
> Note that the stage-II training aims to further improve the performance of the quantized model so that it is better than the full-precision SDv1.5. If reducing the training cost would be the goal, we can remove stage-II training. Stage-I training is able to achieve similar performance as the full-precision SDv1.5 (Fig. 5 of the main paper).
>
> Additionally, we compare training costs between our model and SDv1.5 from scratch (https://huggingface.co/runwayml/stable-diffusion-v1-5). We use the total trained samples to measure the training costs as the training hours depend on the actual hardware used. As in the following table, our stage-I training only requires 0.16% training cost compared with training the SDv1.5, and stage-II training requires 1.6% training cost compared with SDv1.5.
>
> >| Stage | A100 | Batch size | Gradient accumulation | Total batch size | 256x256 iters (K) | 512x512 iterations (K) | Total training samples (M) | Percentage of total training samples compared to SD-v1.5 |
> | :---- | :---- | :---- | :---- | :---- | :---- | :---- | :---- | :---- |
> | Stable Diffusion v1.5 | 256 | 4 | 2 | 2048 | 237 | 1304 | 3155.97 | 100% |
> | Stage-I | 8 | 8 | 4 | 256 | \- | 20 | 5.12 | 0.16% |
> | Stage-II | 32 | 8 | 4 | 1024 | \- | 50 | 51.2 | 1.6% |
>
>
> ---
>
> **Q2.1. Why certain metrics are correlated.**
>
> A2.1. Thanks for the insightful comments! MSE, PSNR, and LPIPS are highly correlated since they measure the changes of image quality. Namely, they calculate the perceptual similarity between two images that align with human perception. More specifically, MSE quantifies the pixel-wise difference between images. Similarly, PSNR, derived from MSE, evaluates the peak error in images via $PSNR = 10 * log_{10}(Max^2 / MSE)$. LPIPS has a different calculation as MSE and PSNR. Yet, it still focuses on significant perceptual differences beyond pixel-level comparisons. Given that perceptual changes often accompany measurable pixel-wise changes, LPIPS correlates with PSNR and MSE.
>
> On the other hand, the CLIP score measures the alignment between the image and prompt, leading to a low correlation with MSE, PSNR, and LPIPS metrics.
>
> ---
>
> **Q2.2. Why quantization affects some layers more than others.**
>
> A2.2. We notice that cross-attention layers and convolutional shortcut layers are more sensitive to low-bit quantization than other layers. Cross-attention layers combine textual and image information. Therefore, any quantization-induced errors for cross-attention layers might greatly change the semantics and content of the generative images, leading to substantial semantic shifts and the loss of important content. For example, in Line 147~149 and Fig. 2 of the main paper, quantizing a cross-attention key layer changes the image content from "a teddy bear" to "a person". Convolutional shortcut layers link different layers and scales of features across the UNet. They facilitate the information flow between layers, enabling the network to learn complex patterns. Quantization of these shortcut connections disrupts the well-established connectivity of features, leading to a breakdown in the coherence of the learned representations (as in Fig. 2).
>
>
> ---
>
> **Q3. Some evaluation datasets, like MS-COCO 2014, may not fully represent the diversity in real-world.**
>
> A3. We agree with the reviewer that most of the datasets used for evaluating text-to-image models have their limitations. In this paper, we evaluate our model on most of the *commonly used benchmark datasets* to get relatively accurate and fair comparisons. Specifically, we use Parti Prompts, TIFA, GenEval scores, and Human evaluation. These benchmark datasets cover a wide variety of prompts and are commonly adopted in evaluating text-to-image models in the literature, like SDXL. Additionally, we would like to kindly clarify that our human preference evaluation includes complex scenes, abstract concepts, and specific artistic styles (as in Fig. 13 and Fig. 14).
>
> Regarding our claim that our quantized model outperforms the full-precision SDv1.5 across all the evaluations, we intend to mention that our BitsFusion can outperform full-precision SDv1.5 across commonly used metrics evaluated in the paper. Based on the feedback from the reviewer, we will modify our writing on Line 61 as our model outperforms the full-precision model *across various evaluation metrics used in the paper*. We hope it can make the paper more rigorous.
>
> We thank the reviewer for suggesting new metrics. We will discuss them in the revised paper. These metrics have not been widely adopted for evaluating text-to-image generation. For instance, Visual Genome is for visual question answering [A], Flickr30k Entities is for evaluating image description [B], Open Images Dataset V4 is for classification, detection, and visual relationship detection [C], and ADE20K is for understanding scene parsing  [E]. Adapting these metrics to evaluate text-to-image models requires a series of experiments to understand how to use them and correlate them with human perception. We take this as a future work.
>
> During the rebuttal, we run the suggested evaluation metric from Conceptual 12M with the CLIP score [D]. By randomly selecting 1K prompts (captions) and using the same CFG scale of 7.5, our 1.99 bits model outperforms full-precision SD-v1.5 (CLIP scores of 0.3084 *vs.* 0.3067).
>
> ---
>
> **Q4. More related methods.**
>
> A4. Thanks for suggesting these papers, which are indeed insightful and relevant! We will discuss these papers in the revised paper.

---

> > ### Author Response · Authors · 2024-08-10
> >
> > Dear Reviewer WzcG,
> >
> > Thank you very much for your insightful feedback and the positive rating. We have included explanations to further clarify the questions. As the deadline for the discussion period is fast approaching, we are eager to see if our explanations meet your expectations and address all your concerns thoroughly. Thank you again for your time and insight.
> >
> > Best regards,
> >
> > The Authors

---

> > > ### Comment · Reviewer_WzcG · 2024-08-13
> > >
> > > Thank you for your detailed response and additional experiments. Your clarifications on training complexity, metric correlation, and dataset evaluation are appreciated. I recommend emphasizing the practical implications for resource-constrained scenarios and exploring why certain layers are more sensitive to quantization. Including the suggested related works will further enhance the paper. I will maintain my current score.

---

> > > > ### Author Response · Authors · 2024-08-14
> > > >
> > > > Thank you for your response and positive rating! We're pleased to hear that your concerns have been addressed. We will incorporate the discussions and related works in the paper.

---

### Author Rebuttal · Authors · 2024-08-06

We thank all reviewers for their valuable suggestions and feedback. We appreciate the reviewers acknowledge the strengths of this paper, including:

- **addressing the challenges** for low-bits quantization of Stable Diffusion (Reviewer W2xs);
- **thorough, comprehensive, and valuable** analysis and observations of the quantization errors for each layer, contributing to the **theoretical understanding** of low-bit quantization for large-scale diffusion models and helpful for **future research** (Reviewer WzcG, Fq1C, tjXw);
- **innovative approaches** (e.g., innovative initialization techniques and improved two-stage pipeline) to compress the UNet to 1.99 bits (Reviewer WzcG, tjXw, W2xs);
- **significant results** for model size reduction and **superior performance** that the quantized model outperforms Stable Diffusion v1.5 across **various metrics and datasets** (Reviewer WzcG, tjXw, W2xs, Fq1C);
the quantized model has **practical potential** on resource-constrained devices and **significant implications** for the industry (Reviewer WzcG, W2xs);
- **a well-organized, well-written, and clear paper**, with experimental details provided, enhancing the paper's **credibility and utility** and ensuring the results can be **reproduced** (Reviewer W2xs, Fq1C, WzcG).

In the following, we provide detailed responses to all concerns from reviewers.

---

### Decision · Program_Chairs · 2024-09-25

**Decision:**

Accept (poster)

**Comment:**

This paper proposes a novel weight quantization method called "BitsFusion" for compressing the Stable Diffusion v1.5 model. The primary goal is to address the issue of large model sizes, which hinder the deployment of diffusion models on resource-constrained devices. The BitsFusion framework quantizes the UNet component of Stable Diffusion v1.5 to 1.99 bits, achieving a 7.9× reduction in model size while improving image generation quality. The results demonstrate the effectiveness of the proposed framework.